# Continuous map of early hematopoietic stem cell differentiation across human lifetime

Hana Komic [1,2,10], Tessa Schmachtel [3,10], Catia Simoes [4,10], Marius Külp [3,5,6,10], Weijia Yu [3], Adrien Jolly [3,6], Malin S. Nilsson [1,7], Carmen Gonzalez [4], Felipe Prosper [4], Halvard Bonig [8], Bruno Paiva [4], Fredrik B. Thorén [1,2] & Michael A. Rieger [3,5,6,9] ✉

Uncovering early gene network changes of human hematopoietic stem cells (HSCs) leading to differentiation induction is of utmost importance for therapeutic manipulation. We employed single cell proteo-transcriptomic sequencing to FACS-enriched bone marrow hematopoietic stem and progenitor cells (HSPCs) from 15 healthy donors. Pseudotime analysis reveals four major differentiation trajectories, which remain consistent upon aging, with an early branching point into megakaryocyte-erythroid progenitors. However, young donors suggest a more productive differentiation from HSPCs to committed progenitors of all lineages. tradeSeq analysis depicts continuous changes in gene expression of HSPC-related genes (*DLK1, ADGRG6*), and provides a roadmap of gene expression at the earliest branching points. We identify CD273/PD-L2 to be highly expressed in a subfraction of immature multipotent HSPCs with enhanced quiescence. Functional experiments confirm the immune-modulatory function of CD273/PD-L2 on HSPCs in regulating T-cell activation and cytokine release. Here, we present a molecular map of early HSPC differentiation across human life.

Hematopoiesis is a continuous regenerative process of cell differentiation, lineage choice and maturation, in which all blood cell lineages arise from a pool of hematopoietic stem cells (HSCs)[1]. Stem cell differentiation is accompanied with a successive loss of self-renewal and multi-lineage potential that is induced by dynamic gene expression networks, resulting in lineage-specific programs and phenotypes.

Despite major advances in the identification of novel surface markers of human HSCs to allow their prospective isolation from different sources, the most immature hematopoietic stem and progenitor cell (HSPCs) fraction remains impure and heterogeneous[2,3]. Taking into consideration the species barriers of xenotransplantation after pre-conditioning regimens as a functional read-out, a purity of approximately 10% of real stem cells with long-term multi-lineage potential, as demonstrated by single cell transplantation, is possible nowadays using the CD49f⁺CD90⁺CD45RA⁻CD34⁺CD38⁻Lin⁻ surface marker profile[4,5]. Furthermore, the heterogeneity and relative contributions of stem cells

¹TIMM Laboratory at Sahlgrenska Center for Cancer Research, University of Gothenburg, Gothenburg, Sweden. ²Department of Medical Biochemistry and Cell Biology, Institute of Biomedicine, Sahlgrenska Academy, University of Gothenburg, Gothenburg, Sweden. ³Department of Medicine 2, Hematology/Oncology, Goethe University Frankfurt, Frankfurt am Main, Germany. ⁴Cancer Center Clínica Universidad de Navarra, Centro de Investigación Médica Aplicada (CIMA), IDISNA, CIBER-ONC number CB16/12/00369 and CB16/12/00489, Pamplona, Spain. ⁵Cardio-Pulmonary-Institute, Frankfurt am Main, Germany. ⁶German Cancer Consortium (DKTK) and German Cancer Research Center (DKFZ), Heidelberg, Germany. ⁷Department of Microbiology and Immunology, Institute of Biomedicine, Sahlgrenska Academy, University of Gothenburg, Gothenburg, Sweden. ⁸Institute for Transfusion Medicine and Immunohematology, Goethe University Frankfurt, Frankfurt am Main, Germany. ⁹Frankfurt Cancer Institute, Frankfurt am Main, Germany. ¹⁰These authors contributed equally: Hana Komic, Tessa Schmachtel, Catia Simoes, Marius Külp. ✉e-mail: m.rieger@em.uni-frankfurt.de

are changing during lifespan, and aging perturbs their physiology and function.

Single cell sequencing has been extensively used in recent years to resolve human bone marrow (BM) hematopoietic cell populations in health and disease[6–14]. These approaches revealed new blood cell subtypes and lineage-specific differentiation stages and identified markers for the prospective isolation of fine-grained progenitor populations[10,11,13]. However, most of these studies did not focus on the most immature HSPC-enriched population due to its rarity in comparison to other BM cell types. Furthermore, immature HSPCs are predominantly quiescent cells with low level of transcription[15,16]. Thus, molecular resolution of the earliest steps of human HSPC differentiation remains limited. A recent study showed that a targeted approach was efficient at detecting low-abundance transcripts while only requiring about one-tenth of the sequencing read depth needed for whole transcriptome analysis, indicating that targeted transcriptomics is a sensitive and cost-efficient alternative when the focus is on interrogating defined transcripts[17].

In this work, we applied a combined Transcriptomic/AbSeq approach to simultaneously quantify the expression of 596 genes at mRNA level and 46 antigens at the protein level on more than 62,000 single cells from FACS-sorted CD34+ HSPCs from 15 donors of different ages. Our results show the heterogeneity among human BM immature HSPCs by zooming into the most immature clusters to uncover molecular traits of the first differentiation decisions in HSPCs. Furthermore, we provide evidence for an immune-modulatory function of CD273/PD-L2 on HSPCs in regulating T-cell responses.

## Results

### Reconstructing a hierarchical organization of CD34+ HSPCs

To resolve cellular heterogeneity and gene expression dynamics in early differentiation steps of human BM HSPCs, we rationally selected 596 genes for deep-targeted Transcriptomic/AbSeq analysis using the BD Rhapsody technology (Supplementary Data 1). First, genes previously reported to be differentially expressed in bulk-purified human immature HSPCs and committed progenitors were selected[18]. Second, we included genes expressed in leukemia stem cells[18–20] and mutated in clonal hematopoiesis and myeloid neoplasms[21]. We further added hematopoietic surface marker genes, immune-modulatory receptors, and cell cycle reporter genes to our panel[8,22]. The 46 oligo-nucleotide-labelled antibodies used to capture the surface proteome of the same cells covered surface markers of immature HSPC subpopulations, oligo- und unipotent progenitors, mature blood cell types and immune receptors, further delineating cell identification and lineage specification (Supplementary Data 2)[4,23,24].

The sequencing saturations were > 91% for mRNA and >70% for AbSeq. In total, 62,277 FACS-sorted CD34+ cells from BM of 15 healthy donors were sequenced at a single cell level and passed all stringent quality control filtering, including 13,000 CD34+CD38− sorted cells (Supplementary Fig. S1). The donors were grouped in three age categories, young adult (n = 5, mean age of 22.4 years), middle-aged (n = 6, mean age of 59.7 years) and old (n = 4, mean age of 75.3 years) (Supplementary Data 3). We generated three separate libraries (mRNA, AbSeqs, sample tags) for sequencing with matched cell barcodes and unique molecular identifiers (UMI) for downstream data integration.

To investigate the CD34+ BM compartment, all cells were utilized for initial clustering analysis. The targeted approach improved sensitivity and accuracy of detecting low-expressed genes. The BD Rhapsody technology allowed improved cell lysis and efficient reverse transcription over droplet-based methods[25]. A particular emphasis was laid on accurate batch-effect correction of the multi-center-generated data (Supplementary Fig. S2). After testing multiple approaches, canonical correlation analysis (CCA) was employed for the mRNA-based analyses[26,27]. Results of mRNA expression-based clustering analysis of all CD34+ cells were projected on UMAP revealing 24 clusters

(Fig. 1a, Supplementary Fig. S3). The expression of genes reported as human HSPC markers (CRHBP, HLF and HOPX) were enriched at the tip of the cluster at the bottom of the UMAP (thereby named HSPC/MPP), suggesting that most immature HSPCs including the HSC fraction reside there (Fig. 1b)[28–31]. This finding was supported by the differential protein expression of known HSPC surface markers CD90, CD34 and the absence of CD45RA and CD38, or other lineage markers (Fig. 1c, Supplementary Fig. S4). All clusters were annotated based on the expression of well-described and previously used marker genes for the respective populations (Fig. 1d). Supplementary Data 4 shows the results of the differential gene expression analysis of each cluster, comparing the cells in one cluster to all other cells in the UMAP (FindAllMarkers algorithm). As some examples, the MKP cluster displayed selective expression of Von Willebrand factor (VWF) and high expression of MPL, the gene encoding for thrombopoietin receptor, and Eo/Baso/Mast cell progenitors were identified by upregulation of HDC, together with high expression of the transcription factor GATA2. To confirm our manual annotations, we utilized unsupervised cell label transfer and a cell label correlation matrix using the dataset from the study of Triana et al. (Supplementary Fig. S5 and Supplementary Data 5)[8]. The most immature HSPCs comprised three clusters that were classified as HSC/MPP, MPP/MK-Ery, and MPP/LMPP. To further address the ontogeny anchor of human hematopoiesis in the represented UMAP, we projected the localization of FACS-sorted CD34+CD38− HSPCs to the UMAP and revealed a high enrichment in the proposed HSPC clusters (Fig. 1e).

In conclusion, the identity of the HSC/MPP subpopulation was clearly depictable by various uncoupled methods representing the cluster with the most immature cells in BM as the anchor point of human hematopoiesis. Our comprehensive UMAP of human HSPCs is compatible with a hierarchical organization of lineage commitment. An early branching point of multipotent HSPCs into the megakaryocytic-erythroid lineage is followed by a predominant production of red blood cell progenitor clusters, as suggested by others.[23]

### Differential gene expression of early immature HSPCs

To focus on the earliest transitions of immature HSPCs into multi- and oligopotent progenitors, we selected the cells that comprised clusters HSC/MPP, MPP/MK-Ery, MPP/LMPP, MEP, MDP-1, GMP-1, GMP-Neut, LyP for a refined unsupervised clustering analysis. Reclustering of 21,395 cells yielded 9 subpopulations, encompassing a sufficient dataset to segregate the most immature stem cell populations (Fig. 2a). We annotated the clusters based on marker gene expression in these populations as HSC-1, HSC-2, multipotent progenitors (MPP), and early committed progenitors (MEP-1, MEP-2, LYP, MDP-1, MDP-2, GMP) (Fig. 2b, Supplementary Data 6). Notably, cluster HSC-1 predominated in this dataset, accounting for 4982 cells. HSC-1 was characterized by the highest expression of well-described stem cell genes (HLF, HOPX, PROM1, CRHBP, MLLT3). Transcriptomic expression levels among the investigated subpopulations revealed distinct mRNA expression patterns for lineage-specific marker genes (Fig. 2b). A comparison of previously published gene signatures for respective HSPC populations confirmed our proposed subpopulation identity[6] (Supplementary Fig. S6). Cells in cluster HSC-1 showed the lowest expression of proliferation and cell cycle-related genes in the notion that the most immature BM HSPCs are quiescent or slowly cycling, as shown for dormant long-term repopulating HSCs (LT-HSCs) in mice[15]. During differentiation, the cells enter the cell cycle and become highly proliferative, as shown by the strong upregulation of MYC and CDK6 (Fig. 2c). However, the results suggest that a lower proportion of immature human HSCs/HSPCs are in a quiescent state than anticipated from data derived from their murine counterparts. This finding is in line with a recent report demonstrating that the majority of human HSCs actively contribute continuously to steady-state hematopoiesis[32]. Hierarchical gating for phenotypic HSPCs based on antigen expression

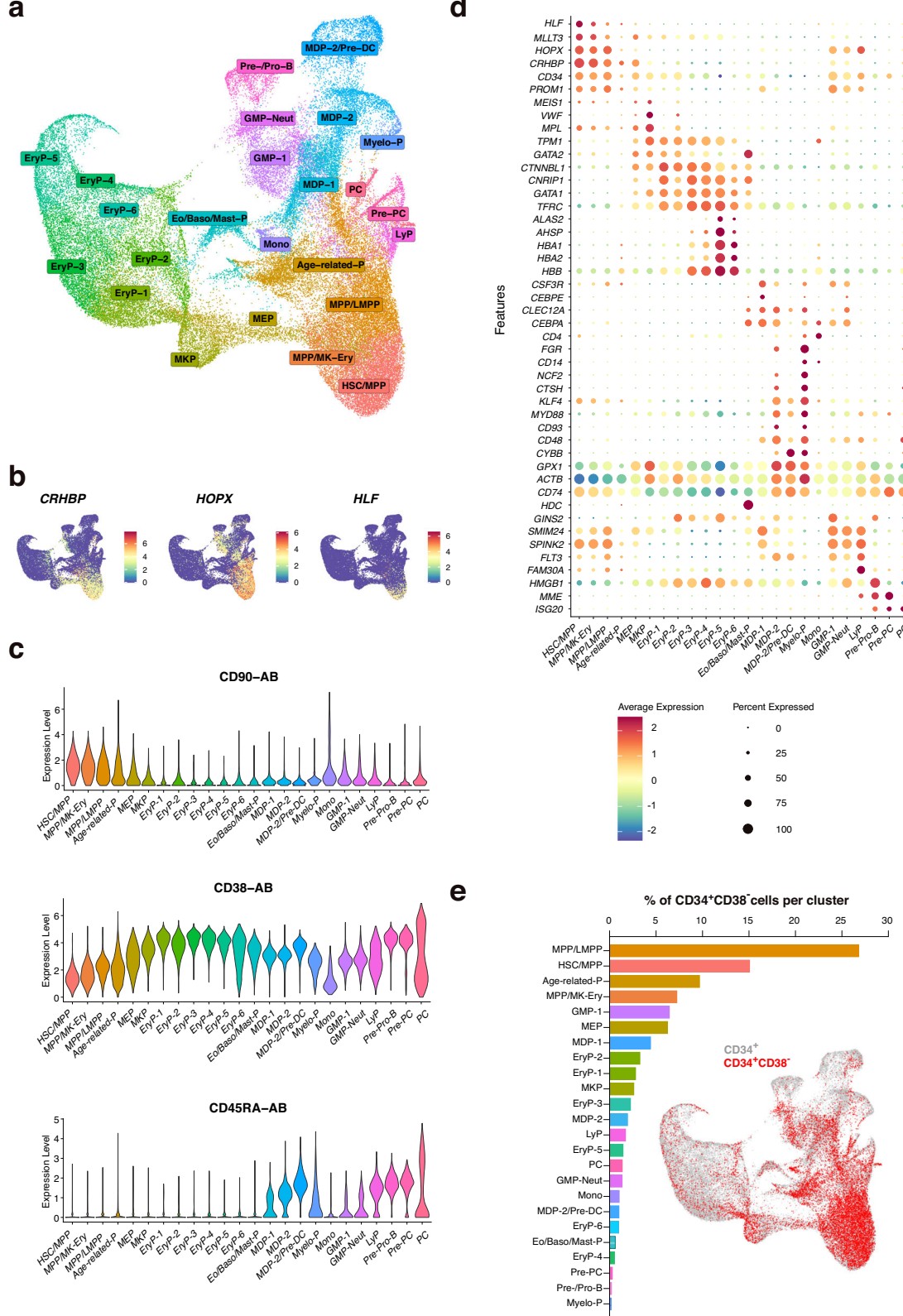

**Fig. 1 | Single cell transcriptomic/AbSeq profiling of CD34+cells from human BM. a** UMAP visualization of gene expression-based clustering analysis of 62,277 CD34+ cells from 15 donors with cluster annotation. **b** Feature plots for the expression of human stem cell gene markers. **c** Violin plots showing the surface protein expression of CD90, CD38 and CD45RA, detected by Transcriptomic/ AbSeq. **d** Dot-plot visualization of gene expression of previously defined cell type-specific markers. **e** Distribution of FACS-sorted CD34+CD38- cells in respective clusters and visualized on the UMAP, demonstrating enrichment in most immature cell clusters.

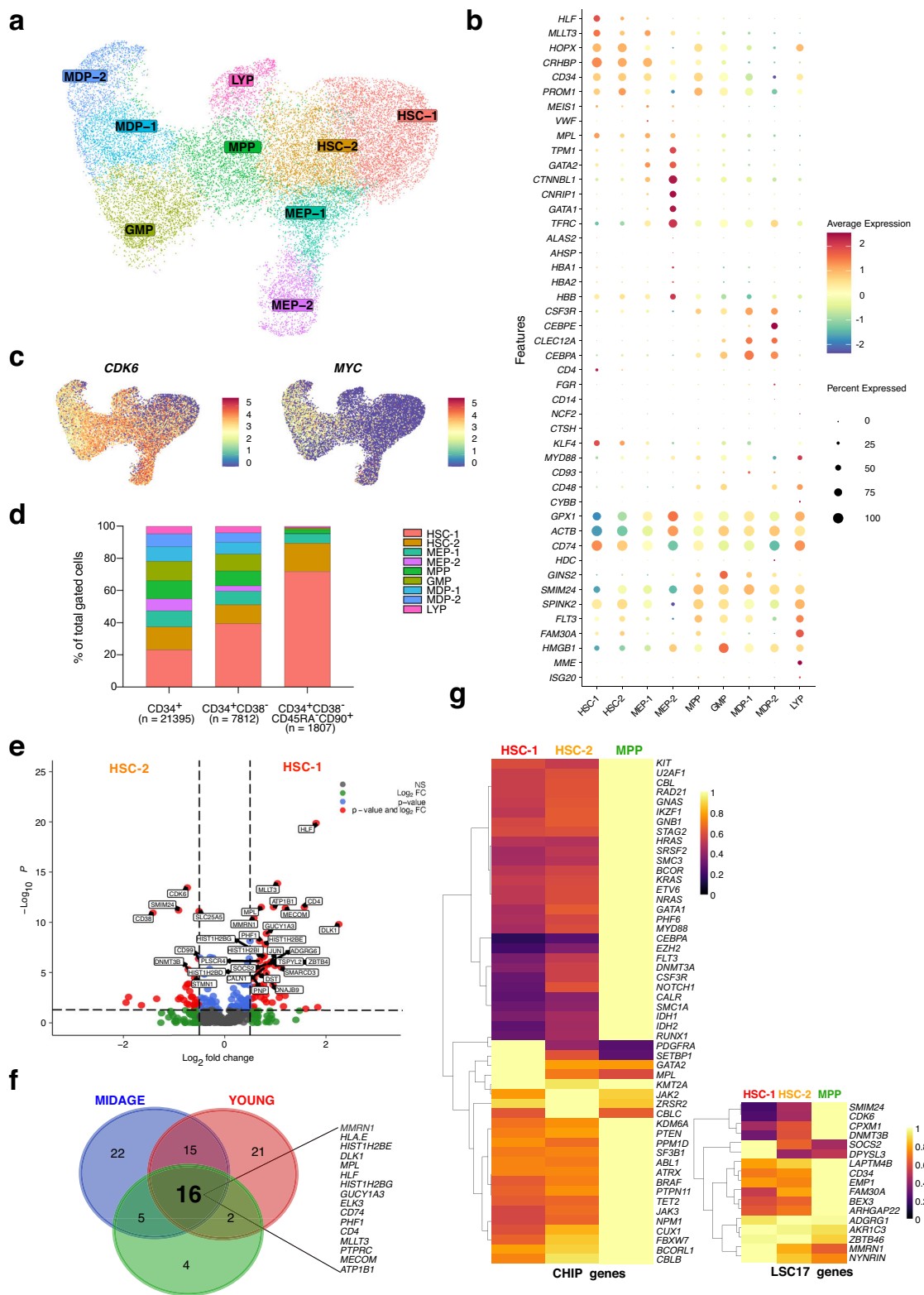

**Fig. 2 | Proteo-transcriptomic profiling and differential gene expression of early immature HSPCs. a** UMAP visualization of gene expression-based re-clustering analysis of cells selected from clusters HSC/MPP, MPP/MK-Ery, MPP/LMPP, MEP, MDP-1, GMP-1, GMP-Neut, LyP (Fig. 1) with cluster annotation. **b** Dot-plot showing expression levels of selected lineage specific genes. **c** Feature plots of cell-cycle and proliferation genes *CDK6* and *MYC*, confirming quiescence of the most immature cells. **d** Hierarchical gating of HSPC populations based on AbSeq surface protein expression and distribution in clusters. **e** Differential pseudobulk gene expression analysis of cells in clusters HSC-1 and HSC-2. Two-sided Wald test with Benjamini-Hochberg correction. **f** Venn-diagram showing comparison of upregulated genes in cluster HSC-1 between individual age group. **g** Clonal hematopoiesis-driver gene and LSC17 stemness gene expression patterns in clusters HSC-1, HSC-2 and MPP.

in our Transcriptomic/AbSeq data confirmed an enrichment of the most immature HSPCs in cluster HSC-1 (Fig. 2d). These results revealed the direction of differentiation of cells from cluster HSC-1 into cells from HSC-2 and allowed for molecular analyses of the most immature human HSPCs. Differential gene expression analyses of HSC-1 and HSC-2 cells using pseudobulk DESeq2 revealed 49 upregulated and 30 downregulated genes in HSC-1 with a fold change >0.5 (log2) (Fig. 2e, Supplementary Data 7). Among these, well-known genes regulating stemness and HSC identity were found, such as *CRHBP*, *MLLT3*, *HLF* and *MECOM*. *MPL* was highly expressed in HSC-1 cells, which is in line with previous reports indicating a role for thrombopoietin signaling for stemness and quiescence[33]. Furthermore, the upregulated *FGD5* gene has previously been used for marking murine LT-HSCs in a transgenic reporter line[34]. Other genes such as *DLK1*, *ADGRG6*, *GUCY1A3*, *KLF4* and *BEX1* were highly expressed in HSC-1 in comparison to HSC-2. We confirmed increased expression of *DLK1* and *ADGRG6* in the most immature HSPC clusters of previously published single cell expression datasets, whereas some HSC-1-specific genes where not differentially detected in these datasets (Supplementary Fig. S7, Supplementary Data 8)[8,13]. Notably, *DLK1*, which plays a critical role in maintaining adult long-term repopulating HSCs in murine hematopoiesis and is essential for stem cell quiescence, was among the most strongly upregulated genes[35]. DLK1 is a non-canonical Notch ligand, which could regulate cell proliferation and differentiation as well as tissue regeneration in Notch-dependent and independent ways[36]. *ADGRG6* encodes for the adhesion G-protein coupled receptor GPR126, that senses mechanical stimuli[37,38]. Interestingly, the expression of this gene is particularly high in MLL-rearranged AML, which belongs to the high risk group of AML[39], but it is unknown to what extent GPR126 impacts on HSPC biology.

Other genes that were found to be upregulated in the HSC-1 cluster have previously not been attributed to HSPC physiology, such as Ankyrin 3 (*ANK3*), Dihydropyrimidinase Like 3 (*DPYSL3*), SWI/SNF Related, Matrix Associated, Actin Dependent Regulator Of Chromatin D3 (*SMARCD3*), and Prostate Cancer Associated Transcript 6 (*PCAT6*). Cells in the HSC-2 cluster showed higher expression of genes involved in cell cycle entry and regulation, such as *CDK6*, *CCNE1* and *MYC*, suggesting that HSC-2 cells begin to re-enter the cell cycle from quiescence (G0 to G1 transition). Along the same lines, *FOXO1*, *SOCS2* and *HLF*, which are associated with quiescence, were more highly expressed in HSC-1 cells. Furthermore, HSC-2 cells expressed early markers of HSPC differentiation, such as *CD38*, *CD48*, *FLT3*, *IL3RA* and *CSF3R* (G-CSF receptor). The epigenetic regulators *EZH2* and *DNMT3B* were upregulated in HSC-2. The expression of *STMN1*, *SMIM24*, *ADA2*, which are highly expressed in lineage-committed progenitors, seemed to be correlated with early differentiation from HSC-1 to HSC-2 cells.

After separating the cells from each age group, the comparison of the differential gene expression in HSC-1 and HSC-2 among age groups using DESeq2 pseudobulk analysis revealed highly overlapping genes, especially for the bona fide stem cell genes in human HSPCs (Supplementary Fig. S8, Supplementary Data 9–11). The analysis identified 16 genes that are upregulated in HSC-1 in all three age groups. However, some genes showed age-dependent specific expression patterns (Fig. 2f, Supplementary Data 12).

Individual somatic mutations in genes associated with myeloid neoplasms in HSPCs can cause the emergence of clonal hematopoiesis (CH)[21]. The mechanism of clonal dominance in HSPCs is dependent on the affected gene and its expression pattern during hematopoietic differentiation[40,41]. Furthermore, the expression of (mutated) leukemia-associated driver genes in leukemia-initiating stem cells and healthy HSPCs has consequences for targeted therapies. Therefore, we included genes associated with CH and myeloid neoplasms in our targeted gene panel and investigated their expression dynamics in cells from clusters HSC-1, HSC-2 and MPP (Fig. 2g). Interestingly, most genes known as CH-drivers were expressed at low level in HSC-1 while

upregulated at the MPP stage. *DNMT3A* - the most frequently mutated gene in CH - was almost undetectable in HSC-1 cells. Most reported *DNMT3A* mutations are loss-of-function mutations suggesting that upregulation of *DNMT3A* is crucial for differentiation induction. Only four genes were highly expressed in HSC-1 and downregulated upon myeloid differentiation. Among these were known stemness genes such as *GATA2* and *MPL*. Next, we determined the expression of genes being part of the clinically relevant LSC17 stemness signature for acute myeloid leukemia (AML)[19]. While most genes were also highly expressed in HSC-1, four genes (*SMIM24*, *CDK6*, *CPXM1* and *DNMT3B*) were almost absent in the most immature cells (Fig. 2g).

## Continuous pseudotime expression of immature HSPCs

To further investigate early changes in gene expression upon differentiation and lineage choice, we performed pseudotime trajectory analysis using Slingshot, which confirmed the differentiation into four lineages: megakaryocytic-erythroid (MEP), lymphoid (LYP), monocytic-dendritic (MDP) and granulocyte-macrophage (GMP). We reanalyzed the Transcriptomic/AbSeq data according to the three age groups of the donors (young, middle-age and old). As expected, Slingshot pseudotime analysis showed no major difference in lineage trajectories of the age groups (Fig. 3a). We then compared the cellular distribution of each subpopulation in the three age groups. While the HSC-1 and 2 clusters of young were smaller than later progenitor subpopulations, the fraction of cells residing in these clusters was markedly increased in BM obtained from middle-age and old donors (Fig. 3b, c). This result confirms previous reports showing that HSC populations expand upon aging[42]. Interestingly, we did not find a reduction of cells in LYP upon aging, which would explain the myeloid bias of hematopoiesis seen in older individuals. On the contrary, we even observed a reduced percentage of GMP and MDP-1 and -2 in BM from aged individuals, in comparison to young donors. Next, we compared the density of cells along pseudotime of the four trajectories (Fig. 3d). The results indicated that cells from young donors show higher commitment rates and a higher overall activity in differentiation and lineage output, while samples from aged donors (both mid and old age groups) show enrichment among most immature cells (Fig. 3d) and a reduction of differentiated progenitors, which was found to be most prevalent in the lymphoid lineage. Last, we compared the expression level of stemness genes upregulated in the subpopulation HSC-1 containing the most immature HSPCs between young and old donors (Fig. 3e). We found a lower mRNA expression of bona fide stemness genes, such as *HLF*, *MECOM*, *CRHBP* and *KLF4* among HSC-1 cells from young donors. The upregulation of these genes with aging was further supported by the fact that middle-age donors displayed intermediate expression of many of these genes. An example is *DLK1*, which was selectively expressed in immature HSPCs and showed a strong age-dependent upregulation.

Due to the deep targeted sequencing of our pre-selected genes, we were able to continuously quantify gene expression during differentiation in all four lineages along pseudotime. Therefore, we applied tradeSeq (trajectory-based differential expression) analysis to our pseudotime-ordered cells[43] and plots of all analyzed genes can be found in the Supplementary Data 13. The continuous gene expression analysis revealed several stage- and lineage-restricted markers with distinctive expression patterns for all four trajectories (Supplementary Fig. S9, Fig. 3f), which warrant future investigations on their function. Genes highly expressed in the most immature HSPCs with a pronounced down-regulation upon differentiation induction were *HLF*, *CRHBP*, *DLK1* and *ADGRG6* (Fig. 3f).

## AbSeq reveals high expression of CD273 on immature HSPCs

The strength of the Transcriptomic/AbSeq approach is the simultaneous detection of mRNA and surface proteome at single cell resolution. Using this technology, potentially new characteristic surface markers and suitable antibodies can directly be identified. We assessed

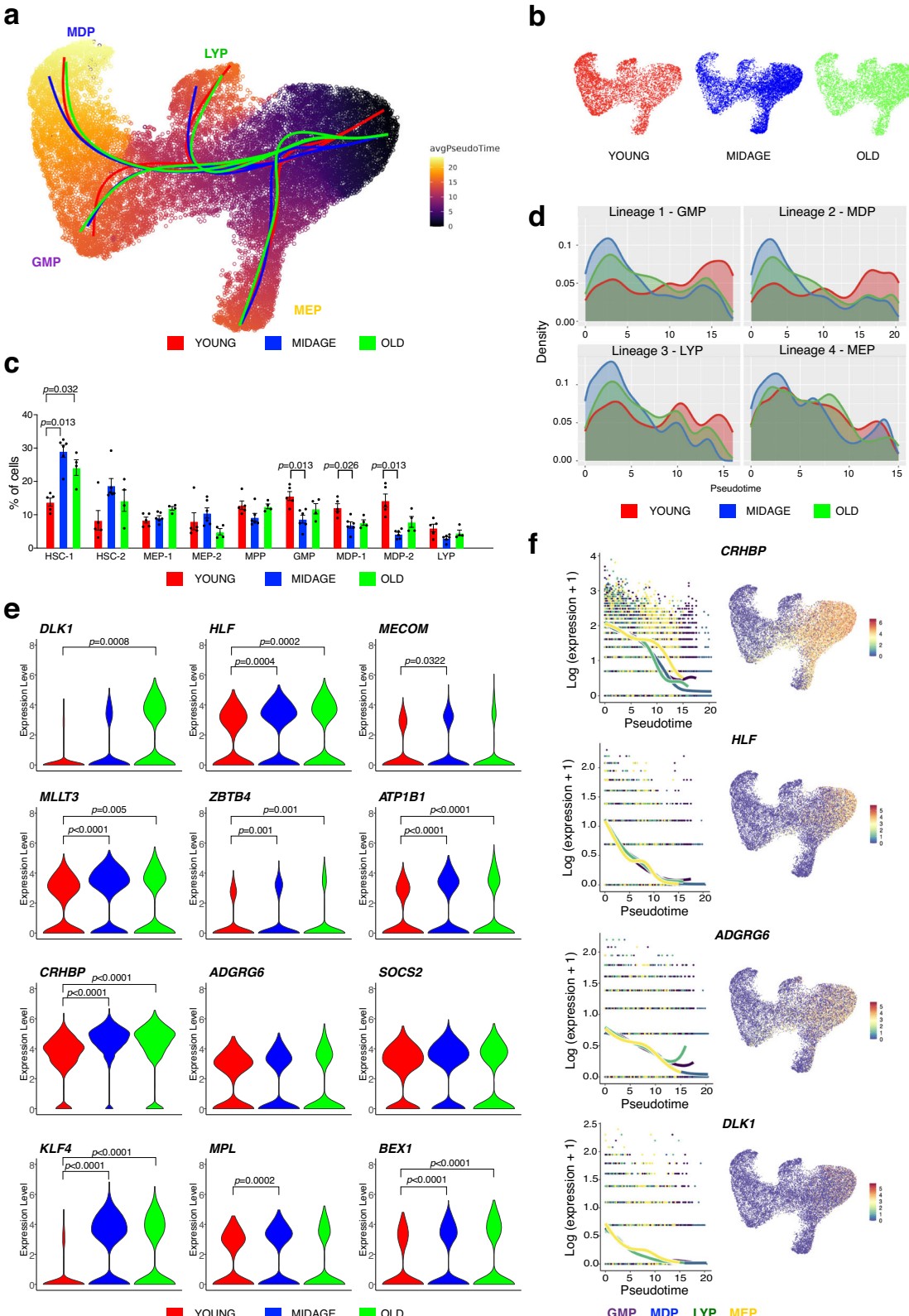

**Fig. 3 | Pseudotime trajectories in different age groups. a** Slingshot pseudotime analysis identified four trajectories. **b** UMAP visualization of cells from each age group. **c** Distribution of cells per cluster from each age group. Wilcoxon Rank Sum test with Holm-Bonferroni correction. Bars represent the mean with SEM. **d** Density plots along pseudotime for each trajectory show age-group specific differences. **e** Violin plots of expression of stem cell-related genes from cells in cluster HSC-1 of each age group. Two-sided Wald test with Benjamini-Hochberg correction. **f** Continuous expression of selected genes highly expressed in HSC-1 along pseudotime calculated by tradeSeq regression model based on lineage trajectories.

the expression level of 46 cell surface proteins, which were selected to cover known markers of HSPC populations. Overall, the annotation of all clusters based on gene expression correlated with the surface marker expression pattern of these subpopulations. The most immature subpopulations HSC-1 and HSC-2 were characterized by high expression of CD34 and CD90 and the absence of early differentiation antigens, CD38 and CD45RA (Fig. 4a, Supplementary Fig. S10). There was a gradual decrease of CD34 and CD90 expression from HSC-1 to MPP, suggesting an early induction of differentiation. Strong upregulation of CD38 and loss of CD90 was found in cells in the MEP-2 cluster. Interestingly, these cells showed high CD123 expression, which is indicative for multipotent and lymphoid progenitors, but not MEPs. Although these cells are committed to the MEP lineage, they still express CD123. MDP-1 and 2 clusters were characterized by strong upregulation of CD371 and CD33. The only significantly upregulated surface markers in HSC-1 as compared to HSC-2 were CD273 and CD62L (Fig. 4b, c, Supplementary Data 14), however, CD62L expression was quite low in both these immature cell clusters. Notably, hierarchical gating based on the surface protein expression for marker-defined HSPC populations showed an enrichment of HSC-1 cluster cells in the fractions with high CD273 expression (Fig. 4d).

To confirm the differential expression of CD273 on the surface of human HSPCs, we performed flow cytometry on BM cells and mobilized peripheral blood HSPCs (Figs. 5, 6a, b, Supplementary Figs. S11 and S12a). We used the well-established surface markers CD90 and CD49f to enrich for immature human HSCs/HSPCs and found this population to display an increased expression of CD273 (Fig. 6a, Supplementary Fig. S12a). Next, we applied hierarchical gating for human HSPCs (CD34⁺CD38⁻CD45RA⁻Lin⁻) and sorted CD273$^{high}$ and CD273$^{low}$ HSPCs (Fig. 5a, b). About 5% of CD34⁺CD38⁻CD45RA⁻Lin⁻ HSPCs express high levels of CD273 in BM and peripheral blood (Fig. 5c). We performed in vitro differentiation cultures with these populations for 7 days and measured the percentage of immature HSPCs by flow cytometry. These experiments showed that the CD273$^{high}$ HSPC population displayed a higher percentage of immature HSPCs after 7 days of culture, as determined by markers CD34⁺CD90⁺ and CD34⁺CD133⁺, and a higher median expression of Endothelial protein C receptor (EPCR) (Fig. 6b, Supplementary Fig. S12b). Next, we determined the expression of stem cell-associated gene products by sensitive capillary Western blots (Simple Western) in CD273$^{high}$ and CD273$^{low}$ HSPCs (Fig. 6c). Protein detection by Simple Western confirmed the differential expression of CD273 in these populations. Furthermore, CD273$^{high}$ HSPCs expressed higher levels of HLF and ATP1B1 than CD273$^{low}$ HSPCs, which is in accordance with the mRNA expression results seen for cells in clusters HSC-1 versus HSC-2 as described above. Interestingly, the early induced cell cycle-related factor CDK6, which marks cells entering the cell cycle, was significantly lower expressed in CD273$^{high}$ HSPCs[6]. These results suggest that CD273$^{high}$ cells represent a more quiescent population of human HSPCs. Next, we performed RNA-Seq on FACS-sorted CD273$^{high}$ and CD273$^{low}$ HSPCs to reveal differentially expressed genes among these subpopulations (Fig. 6d). These experiments demonstrated increased expression levels of stemness genes, such as *Thy1, HOPX, DLK1, MPL*, and *MLLT3* in CD273$^{high}$ HSPCs, while *CDK6* was downregulated in CD273$^{high}$ cells, suggesting a more pronounced quiescence in CD273$^{high}$ HSPCs. These results were highly concordant with the gene expression measured in our Transcriptomic/AbSeq data set, using a similar hierarchical gating strategy for CD273$^{high}$ HSPCs (Fig. 6e, Supplementary Data 15). Finally, to assess whether CD273$^{high}$ HSPCs were more quiescent than CD273$^{low}$ HSPCs, we performed video-microscopy-based single HSPC tracking to quantify the entry into first cell division after FACS-isolation, as an indication of how deep cells are in the G0 phase (Fig. 6f). Indeed, these experiments demonstrated that CD273$^{high}$ HSPCs displayed a delayed entry into cell cycle and first division as compared with their CD273$^{low}$ counterparts. Along these lines,

CD273$^{high}$ HSPCs in vitro culture demonstrated a much slower expansion over three days in comparison to CD273$^{low}$ HSPCs (Fig. 6g). Last, we performed colony-forming unit (CFU) assays with serial replating to demonstrate multi-lineage differentiation potential. We compared CD273$^{high}$ and CD273$^{low}$ HSPCs with CD90⁺CD49f⁺ HSPCs. We did not observe any difference in CFU ability and numbers, myeloid lineage differentiation or serial replating capacity between these populations, demonstrating that CD273$^{high}$ HSPCs have equal – but not superior in vitro capacity as CD90⁺CD49f⁺ HSPCs (Fig. 6h, i, Supplementary Fig. S13)[4].

## CD273 functions as an immune-modulatory receptor in HSPCs

CD273 (PD-L2) binds to the receptor PD-1 on T-cells, known as an important immune checkpoint, to prevent excessive T-cell activation[44]. To investigate the potential immune-modulatory role of PD-L2 expression in the early HSPC compartment, a mixed lymphocyte reaction (MLR) assay was established. CD34⁺ HSPCs and T-cells from four healthy human donors were MACS-enriched. In average, about 1% of HSPCs expressed CD273 on their surface (Fig. 5b, c). Donor-individual HSPCs were co-cultured with unmatched T-cells isolated from another donor, leading to individual allogeneic pairs. T-cells were stimulated with beads coated with antibodies against CD2, CD3, and CD28, which trigger robust T-cell activation. The HSPCs were pre-treated either with a neutralizing CD273/PD-L2 antibody or with an IgG isotype control to evaluate the specific contribution of CD273/PD-L2 to immune modulation of T-cells. The co-cultures were then compared to control conditions, where T-cells were cultured separately.

To monitor T-cell proliferation dynamics in distinct T-cell subtypes over time during co-culture, in the presence and absence of CD273/PD-L2 neutralization, CSFE (CellTrace) and T-cell surface marker expression were determined by flow cytometry (Fig. 7a, Supplementary Fig. S14) and quantified using the proliferation index (Fig. 7b). The co-culture with HSPCs suppressed the proliferation of CD8⁺ T-cells in comparison to T-cells alone. However, increased CD8⁺ T-cell proliferation was effectively rescued by blocking CD273/PD-L2. This observation suggests that HSPCs suppress CD8⁺ T-cell proliferation through PD-L2 interactions. In contrast, the proliferation of CD4⁺ T-cells was not altered by the MLR itself, indicating that CD4⁺ T-cells might not be as sensitive to HSPC-mediated suppression as CD8⁺ T-cells under these conditions. However, when CD273/PD-L2 was blocked, there was a marked increase in CD4⁺ T-cell proliferation. This suggests a delicate balance between T-cell stimulation and suppression within the co-culture environment, and that blocking CD273/PD-L2 may tip this balance towards stronger CD4⁺ T-cell activation.

To further determine the activation status of CD8⁺ T-cells, the expression of CD69, a marker of early T-cell activation[45], was assessed (Fig. 7c). The co-culture showed a modestly non-significantly increased CD69 expression on CD8⁺ T-cells (Fig. 7d). By contrast, CD273/PD-L2 inhibition resulted in an elevation of CD69 levels, reinforcing the idea that PD-L2 mediates a key suppressive signal for CD8⁺ T-cell activation.

Next, the effects of HSPC co-culture on memory T-cell populations were explored using flow cytometry (Fig. 7e). The data showed a relative decrease in naïve T-cell ($T_N$) and central memory T-cell ($T_{CM}$) populations, while effector memory T-cells ($T_{EM}$) increased upon MLR, confirming that contact with HSPCs promotes T-cell specification towards more activated states. Of note, CD273/PD-L2 blockade resulted in a notable reduction of the fraction of CD4⁺CD25$^{hi}$FOXP3⁺ regulatory T-cells ($T_{regs}$). This result points to CD273/PD-L2 playing a dual role in immune suppression: (1) direct inhibition of T-cell activation through receptor-ligand interactions, and (2) indirect suppression via the induction and expansion of $T_{regs}$.

To substantiate phenotypic changes in T-cell subsets, T-cell function was evaluated by analyzing cytokine levels in the supernatants of the co-cultures using multi-cytokine assays (Fig. 7f,

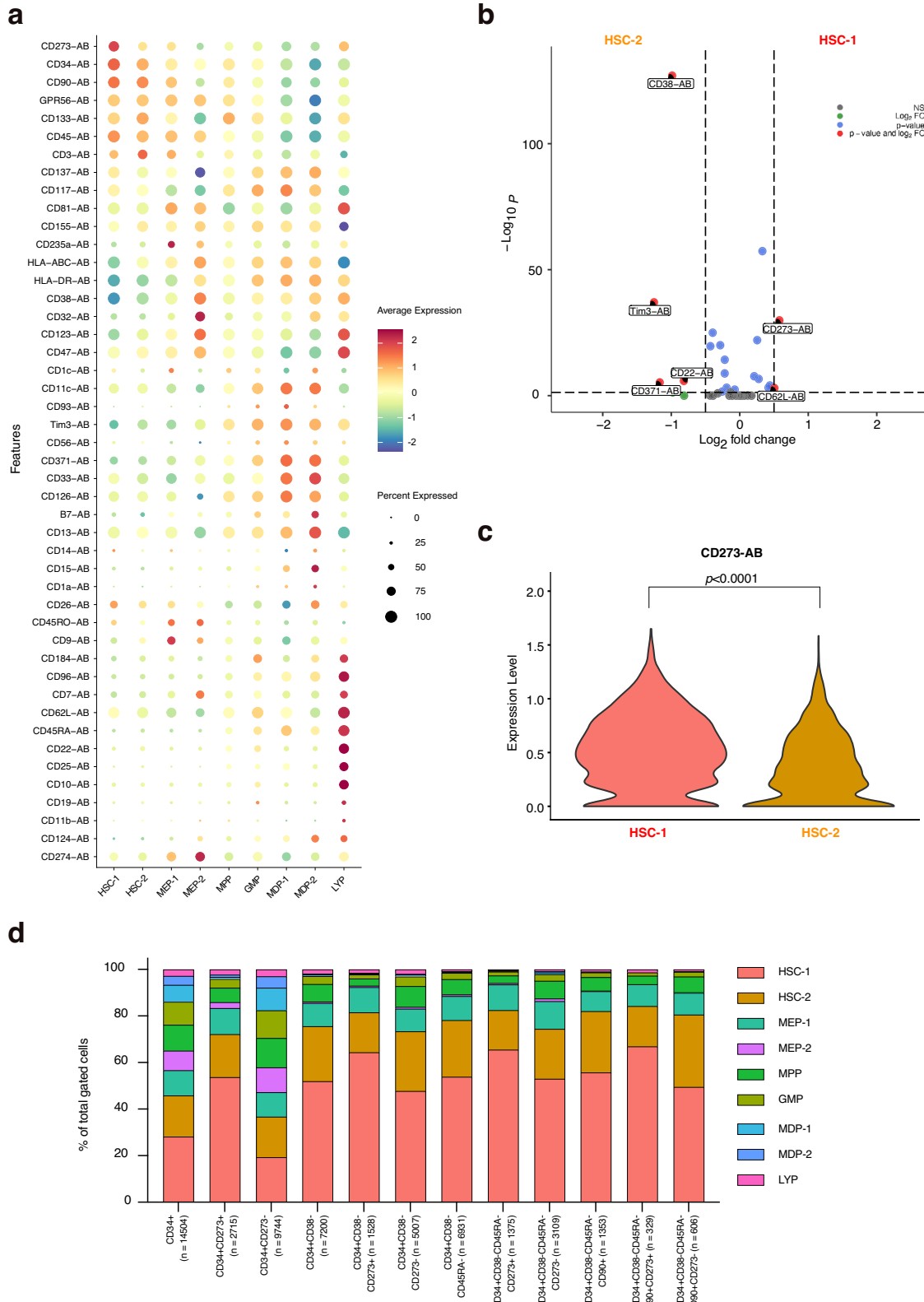

**Fig. 4 | Surface protein expression in immature HSPC subpopulations determined by AbSeq. a** Dot-plot showing surface expression level of all analyzed proteins. **b** Volcano plot represents differentially expressed surface proteins between cluster HSC-1 versus HSC-2. Wilcoxon Rank Sum test with Bonferroni correction. **c** Surface expression of CD273 (PD-L2) in cells from clusters HSC-1 and HSC-2. Wilcoxon Rank Sum test with Bonferroni correction. **d** Hierarchical gating of HSPC populations based on the AbSeq surface protein expression and distribution in clusters.

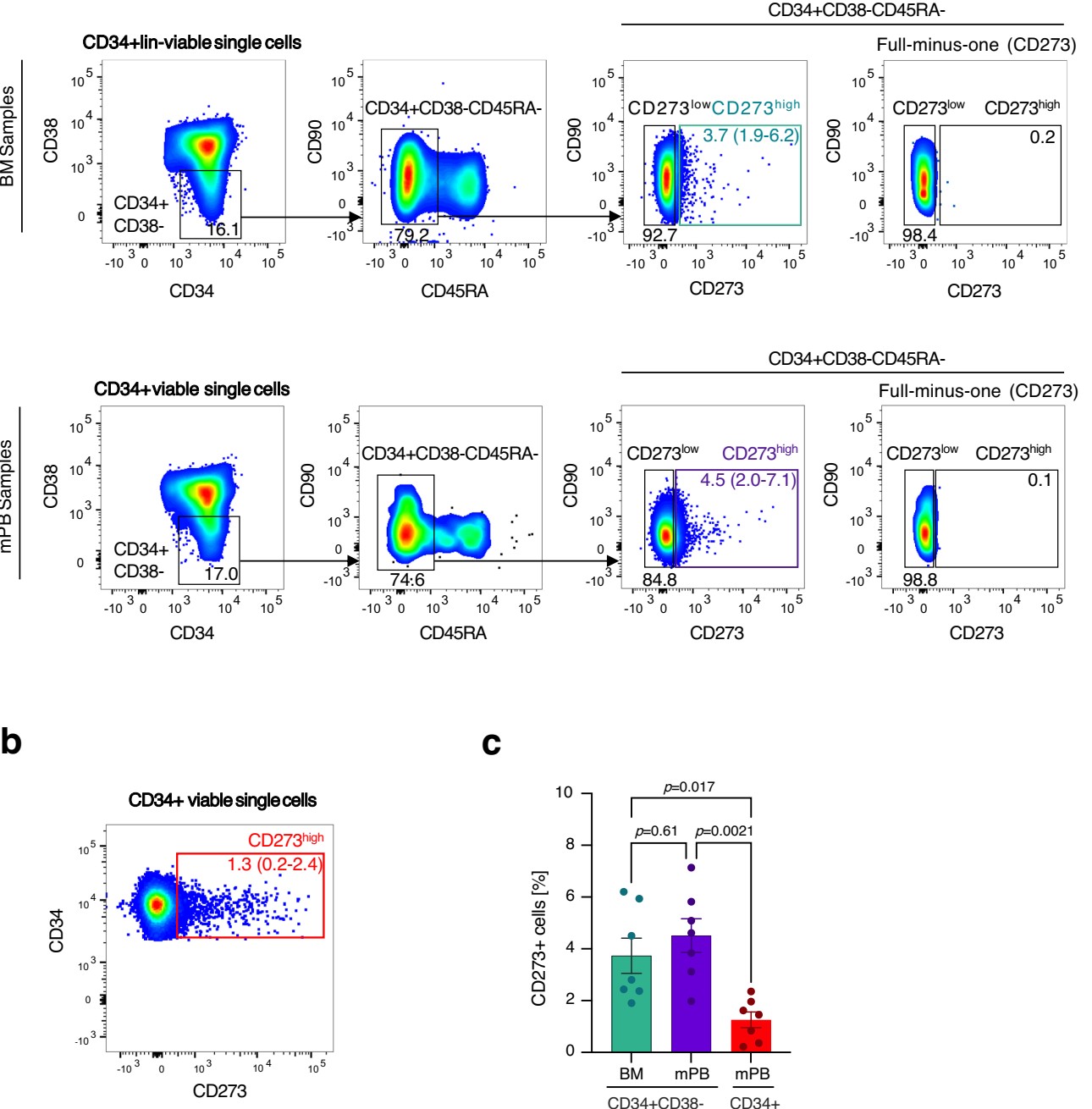

**Fig. 5 | FACS sorting of CD273[high] and CD273[low] HSPCs. a** Pseudocolor density plots illustrating the hierarchical gating strategy used for isolation of CD34[+]CD38[−]CD45RA[−]lin[−] HSPCs according to their CD273 expression from adult bone marrow (BM) and mobilized peripheral blood (mPB). The first plot shows pre-gated viable, singlet CD34[+] lin[−] cells. **b** CD273 expression of CD34[+] MACS-enriched mPB. **c** Percentage of CD273[high] cells within the indicated populations. Bars represent the mean with SEM (*n*=7 donors). One-way ANOVA with Tukey's multiple comparisons test.

Supplementary Fig. S15). The T-cell activation cytokines IFN-γ, TNF-α and IL-6 were elevated in CD273/PD-L2 neutralized co-cultures compared to T-cell mono-cultures. The co-culture of T-cells with HSPCs decreased the secretion of the pro-inflammatory cytokine IL-17A, while CD273/PD-L2 blocking markedly elevated IL-17A levels. These findings strengthen the hypothesis that the interaction of CD273/PD-L2 promotes an immunosuppressive environment, leading to reduced T-cell activation and inflammation.

## Discussion

In this study, we investigated molecular cues associated with human HSPC identity, early differentiation induction and lineage choice using single cell proteo-transcriptomic sequencing. The targeted sequencing approach allowed accurate detection of weakly expressed genes at single cell resolution of CD34[+] BM cells across different age groups for continuous gene expression quantification. The combination of surface marker detection with the single cell transcriptome enabled

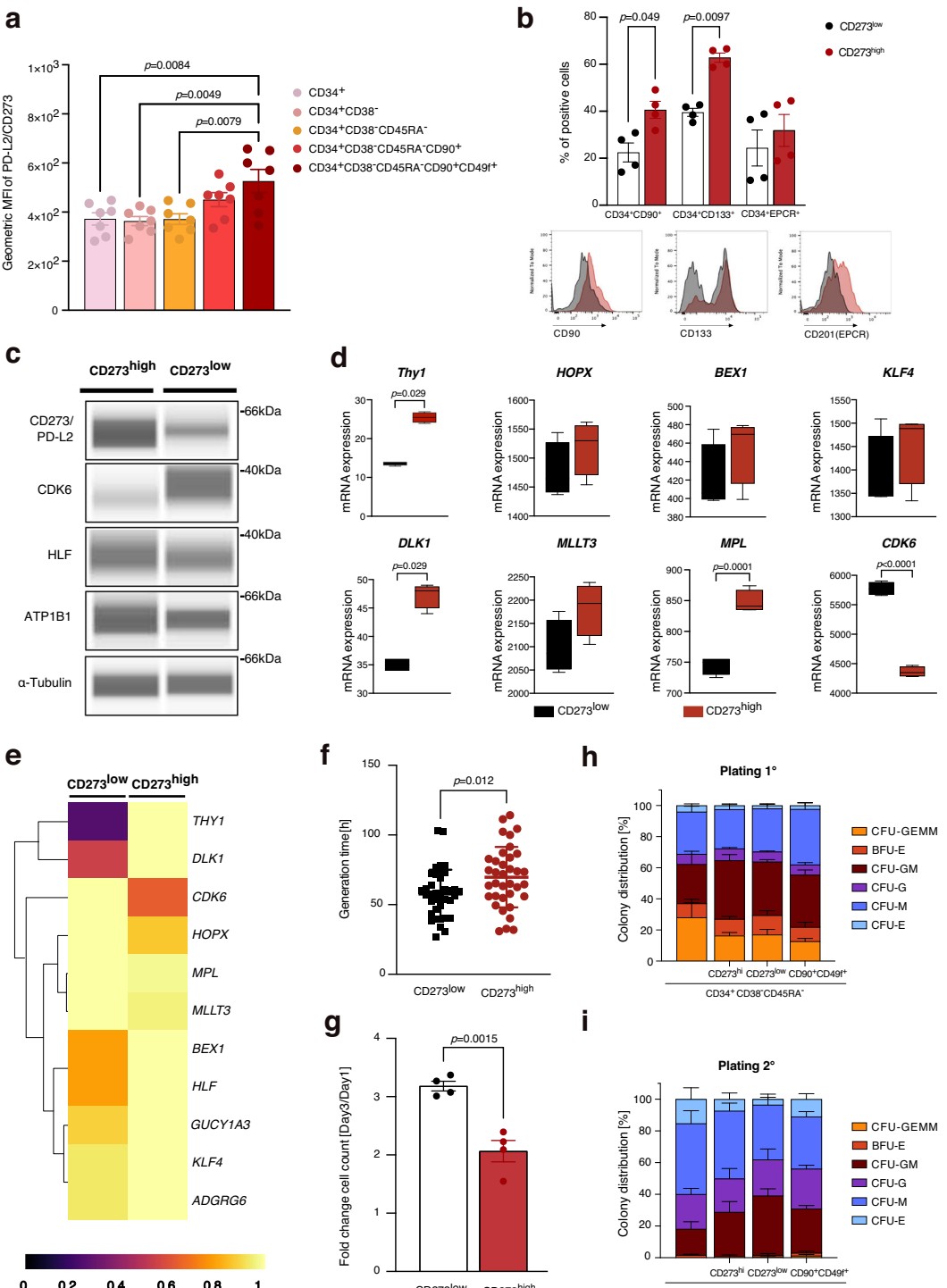

**Fig. 6 | Functional characterization of CD273/PD-L2 expressing HSPCs. a** CD273 surface expression level on different HSPC populations (*n* = 7 donors) by flow cytometry. One-way ANOVA with Tukey's multiple comparisons test. **b–f** CD273high and CD273low CD34+CD38−CD45RA− HSPCs were FACS-sorted for analyses (see Fig. 5). **b** CD273high HSPCs show delayed in vitro differentiation. Cells were analyzed after 7 days of culture by flow cytometry (*n* = 4 donors). Two-way ANOVA with Šídák's multiple comparisons test c) Protein expression of CD273 and stem cell-related marker proteins in FACS-sorted CD273high and CD273low HSPCs determined by Simple Western technology. **d** Expression of stem cell signature genes measured by bulk RNA-Seq of FACS-sorted CD273high and CD273low HSPCs (*n* = 4 donors). Box extends from 25th–75th percentiles, marking the mean with a distinct line. The whiskers reach out to the maximum and minimum data point. Two-tailed Mann-

Whitney test (*THY1, BEX1, KLF4, DLK1*) and two-tailed unpaired t-test (*HOPX, MLLT3, MPL, CDK6*). **e** Expression of stem cell signature genes measured on hierarchically gated CD273high and CD273low HSPCs from AbSeq analysis showed the same pattern as bulk RNA-Seq in d. **f** Videomicroscopy-based single cell tracking of FACS-sorted CD273high and CD273low HSPCs showed delayed entry into first cell cycle of CD273high HSPCs (*n* > 35 cells). Two-tailed *t*-test. **g** Fold expansion of FACS-sorted CD273high and CD273low HSPCs after 3 days in vitro culture (*n* = 4 donors). Two-tailed unpaired *t*-test. **h** Colonies determined by colony-forming unit assay (CFU) after 14 days (*n* = 5 donors). Two-way ANOVA with Tukey's multiple comparisons test. **i** Replating of primary CFU (*n* = 4 donors). Two-way ANOVA with Tukey's multiple comparisons test. Bars represent the mean with SEM.

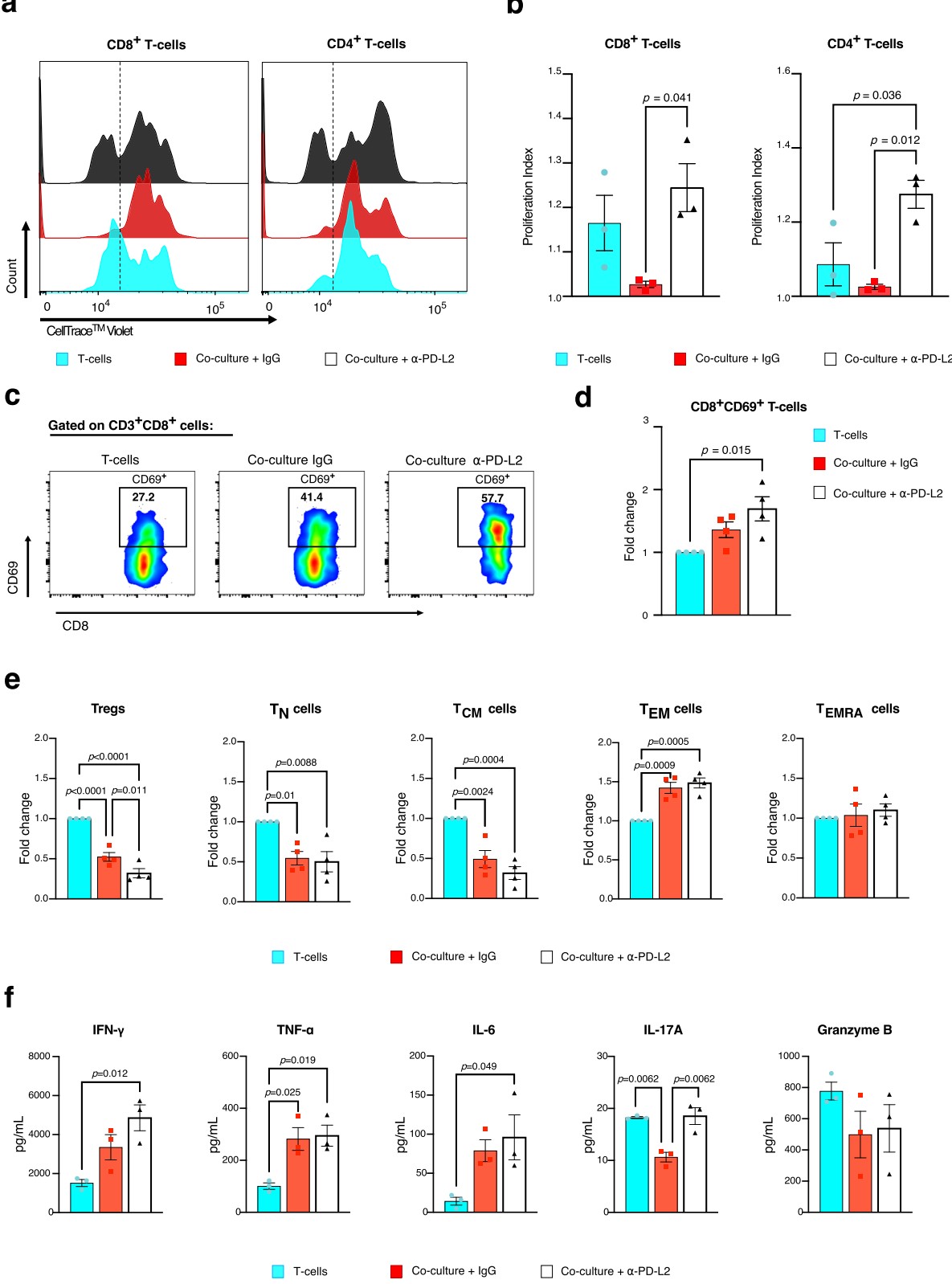

**Fig. 7 | Immune-modulatory function of CD273/PD-L2 on HSPCs.** T-cells were co-cultured with allogeneic HSPCs in the presence of either PD-L2 blocking antibody or an isotype control antibody for 72 h, and compared to a T-cell mono-culture. **a** Proliferation of CD4⁺ and CD8⁺ T-cells by CFSE labeling (*n* = 3 donors). **b** Proliferation index of CD4⁺ and CD8⁺ T-cells (*n* = 3 donors). One-way ANOVA with Tukey's multiple comparisons test. **c** Flow cytometry analysis of CD69 expression on CD8⁺ T-cells. **d** Activation of CD8⁺ T-cells (*n* = 4 donors). One-way ANOVA with Holm-Šídák's multiple comparisons test. **e** T-cell subtype abundances relative to T-cell mono-culture (*n* = 4 donors). T$_{regs}$ regulatory T-cells; T$_N$ naïve T-cells, T$_{CM}$ central memory T-cells; T$_{EM}$ effector memory T-cells; T$_{EMRA}$ terminally differentiated effector memory T-cells re-expressing CD45RA. One-way ANOVA with Holm-Šídák's multiple comparisons test. **f** Cytokine-bead array assay of co-culture and T-cell mono-culture supernatants (*n* = 3 donors). One-way ANOVA with Tukey's multiple comparisons test. All bars represent the mean with SEM.

direct detection of new surface markers with differential expression in the most immature HSPC fraction.

Thereby, we identified CD273/PD-L2 as a differentially expressed surface marker in human HSPCs. CD273 is a ligand of PD-1 and plays an important role as an immune-modulatory molecule. The inhibition of the PD-1/PD-L1/L2 axis is highly relevant in cancer treatment, known as immune checkpoint blockade[46]. Activated PD-1-expressing T-cells are suppressed by the interaction with PD-L1 or L2, which are commonly overexpressed on cancer cells or myeloid cells in many solid malignancies[47]. In murine hematopoiesis, the expression of CD274 (PD-L1) was reported to allow murine HSPCs to escape immune reactions mediated by HSPC-reactive T-cells[48,49]. HSPCs have been proposed to have immune-modulatory functions due to expression of MHC class II molecules that may allow HSPCs to present antigens influencing T-cell responses[50]. Lately, Beta-2-microglobulin expression on HSPCs has been proposed as a crucial "don't eat-me" signal in zebrafish, regulating the HSPC-macrophage interaction and HSPC clonality[51]. However, the expression of PD-1 ligands may serve other purposes than protecting HSPCs against auto-reactive T-cells. In our study, we could confirm a higher surface expression of CD273/PD-L2 on a subset of human immature HSPCs by flow cytometry. Prospectively isolated HSPCs with high CD273 expression were found to display delayed differentiation, multi-lineage potential and enhanced quiescence, all attributes important for HSPC physiology. Increased expression of stemness genes suggests that potent human HSPCs may reside in the CD273[high] HSPC fraction, which needs adequate functional in vivo assessment. Here we demonstrated an immune-modulatory function of CD273/PD-L2 expressed by HSPCs. Allogeneic T-cell proliferation and activation were elevated upon CD273/PD-L2 blockade. Furthermore, the release of pro-inflammatory cytokines increased. Lastly, T-cell differentiation into regulatory T-cells (T$_{reg}$) was reduced after CD273/PD-L2 inhibition on HSPCs. These results suggest that CD273/PD-L2 plays an immunosuppressive role on HSPCs, preventing excessive T-cell activation and potential auto-reactivity. The data highlight CD273/PD-L2 as a safeguard for immature HSPC populations, acting as both a direct inhibitor of T-cell responses and as a facilitator of T$_{reg}$-mediated immune regulation. The presence of regulatory T-cells within the HSPC BM niche may provide a privileged environment for HSPC physiology[52,53]. These findings provide valuable insights into the complex immune interactions between HSPCs and T-cells, and underscore the importance of CD273/PD-L2 in maintaining immune homeostasis.

In agreement with recent reports of single cell sequencing studies using human BM cells and cord blood cells, we here present a structure of a hierarchical organization of human hematopoiesis[11,32,54]. Initial single cell sequencing data of human BM suggested that there is a "cloud" of HSPCs, which very rapidly become committed to a distinct lineage[9]. Our data provide evidence for a model of stepwise commitment and restriction to distinct lineages, which would be compatible with decades-long findings of the murine hematopoietic hierarchy. There was a considerable delay in pseudotime, before the most immature HSPCs at the anchor point of the trajectories diverged into the first lineage-committed progenitors (Mek/E lineage) suggesting that multipotency is maintained after the first steps of early differentiation. While we have molecular evidence that most immature HSPCs are multipotent, rather than a heterogeneous mixture of lineage-restricted HSPCs, we did not perform single cell transplantations to consolidate their multipotency. Pseudotime and tradeSeq analyses suggested a balanced and productive differentiation of HSPCs from young donors, and a delay in differentiation of all lineages in mid-age and old donors by an increased expression of stemness genes. This diminished differentiation was most pronounced in the lymphoid commitment. These findings suggest that there is an age-dependent molecular regulation of progenitor production by early events in HSPC

differentiation, contributing to the increased frequency of immature HSPCs as well as the reduced lymphopoiesis and reduced normal range of hemoglobin seen in old people. Whether these changes were caused intrinsically or orchestrated by the aged environment requires further investigation.

A major strength of our study is the accurate quantification of even lowly expressed genes at single cell resolution by targeted sequencing with a low individual dropout rate, which distinguishes our study from previous attempts. However, this strength naturally comes with the tradeoff that our analyses were restricted to a set of pre-selected protein-coding genes. Nevertheless, our data demonstrates that the selection of these genes was sufficient to clearly resolve human HSPCs populations with similar efficiency as compared to whole transcriptome analysis.

In conclusion, here we present a rich data set of single cell gene expression changes influencing the earliest steps of human HSPC differentiation. Using our approach, we discovered that the immune-modulatory molecule CD273/PD-L2 is highly expressed on the most immature HSPCs and its functional capacity in suppressing allogeneic T-cell proliferation, activation and pro-inflammatory cytokine release. These findings have implications on our understanding of human HSPC physiology and differentiation control, important features for therapeutic manipulation and expansion of human HSPCs.

## Methods

### Subjects

Our research complies with all relevant ethical regulations. BM samples and G-CSF mobilized apheresis product samples were collected from healthy donors at Sahlgrenska University Hospital (Gothenburg, Sweden), at Clínica Universidad de Navarra (Pamplona, Spain) and at University Hospital Frankfurt (Frankfurt am Main, Germany). The sample collection was approved by the ethics committee (ethical permits 011-17 Gothenburg, #329/10 Frankfurt and 2019.143 Pamplona) and conducted according to the principles of the Declaration of Helsinki. All subjects gave written informed consent prior to sampling. We included 15 BM samples in our study, five of young age (20–23 years old), six of middle age (52–65 years old) and four of old age (70–84 years old). In total, there were eight male and seven female participants, with equal distribution in age groups (Supplementary Data 3). The prospective isolation of CD273[high] and CD273[low] HSPCs was performed using G-CSF mobilized apheresis product samples from 33 healthy donors. Donor distribution included 27 males and 6 females (Supplementary Data 16). Epidemiology was consistent with the typical demographics of a German donor registry[55]. No sex-dependent sub-analysis has been performed in this study.

### BM sample preparation

BM samples were processed immediately upon the collection. Cells of interest were isolated by density gradient centrifugation with Lymphoprep (Stemcell Technologies) or Ficoll-Paque Plus (GE HealthCare) according to the manufacturer's instructions. Cells were either employed immediately or cryopreserved for downstream analyses.

Transcriptomic/AbSeq workflow required cell resuspension in BD Stain Buffer (BD Biosciences), followed by staining with oligonucleotide-conjugated antibodies AbSeq (Supplementary Data 2) and Sample Tags, and fluorochrome-conjugated antibodies: anti-CD14-PE-Cy7, anti-CD34-BV421 and anti-CD38-BV510 and unconjugated HLA-ABC (BD Biosciences) (Supplementary Data 17). Cells were stained for 45 min on ice, and then washed twice with BD Stain Buffer. CD34$^+$ precursors from young age group samples were sorted using a BD FACSAria II (BD Biosciences) based on the expression of CD34, FSC and SSC. After sorting, cells were stained with sample tags, and sequentially stained with a master mix of AbSeq.

To enrich for the most immature compartment of BM, cells from middle and old age group samples were divided into two fractions and labeled with different Sample Tags (BD Biosciences). Stained cells were sorted using a BD FACSAria III flow cytometer (BD Biosciences). Anti-CD14 mAb was included to exclude CD14$^+$ cells and to enhance the resolution between CD34$^+$ and CD34$^-$ cells. One fraction of each sample was used for isolating CD34$^+$ cells, while the other fraction was used for sorting CD34$^+$CD38$^-$ cells. Upon collection, the two cell fractions were washed and pooled together.

## Single cell capture

Isolated cells were stained with Calcein (ThermoFisher Scientific) and Draq7 (BD Biosciences) to assess cell concentration and viability prior to single cell capture. BD Rhapsody Express Single Cell Analysis System (BD Biosciences) nanowell cartridges were primed, and cells were loaded (maximum two subjects per cartridge) and incubated at room temperature. Cell Capture Beads (BD Biosciences) were mixed by pipetting and loaded onto the cartridge. Captured cells were lysed and beads with bound mRNA, AbSeqs and Sample sample tags were pooled into one tube. Reverse transcription and Exonuclease I treatment were performed according to the manufacturer's protocol (BD Biosciences).

## Library preparation and sequencing

mRNA, Sample Tag, and AbSeq libraries were generated using BD Rhapsody Targeted mRNA, AbSeq Amplification and BD Single-Cell Multiplexing Kits and protocols (BD Biosciences). The custom gene panel employed in the study comprised 596 carefully selected genes for hematopoietic and progenitor cells, leukemia stem cells, clonal hematopoiesis-related genes and for cell surface and cell cycle markers (Supplementary Data 1). The final libraries were sequenced with 20% PhiX control DNA spiked to increase sequence complexity. Sequencing was done using NovaSeq S1, 2 × 50 reads (Illumina), NextSeq 550 and NextSeq 500, 2 × 75 reads with custom set up 60 - 8 - 0 - 42 (Read1- i7 index- 0 - Read2) for all sequencers. The sequencing was performed at National Genomics Infrastructure (NGI) Sweden/Science for Life Laboratory (Stockholm, Sweden), Max Planck Institute for heart and lung research (Bad Nauheim, Germany) and Hemato-Oncology Department at CIMA-University of Navarra (Navarra, Spain), respectively. The number of allocated reads per cell was 10,000–17,000 per mRNA, 200-500 per each AbSeq and 120–600 per sample tag library.

## Data analysis

Single cell proteo-transcriptomic fastq files from all cartridges were processed separately using the BD Rhapsody Targeted Analysis Pipeline (v1.10.1 and 1.10, respectively) on the Seven Bridges Genomics platform (https://www.sevenbridges.com). The pipeline generated single-cell gene and protein expression matrices after RSEC (Recursive Substitution Error Correction) UMI adjustment. Transcriptomic/AbSeq data analysis was performed in R (v. 4.2.0) using the Seurat package (v. 4.3.0)[56]. The expression matrices were checked for UMI and feature count distribution, and samples with low UMI/feature counts were removed. High quality singlets were defined as putative cells if > 75% of Sample Tag reads came from a single tag, with counts for other tags considered as Sample Tag noise. To recover more putative cells that were initially not recognized as high quality, the algorithm subtracts the expected number of noise counts per each cells from each Sample Tag. Cells that showed counts of two or more Sample Tags exceeding minimum threshold were labeled as multiplets, clearly indicating the presence of >1 cell per microwell, while remaining cells are recognized as undetermined.

## Batch effect correction and data normalization

Cells from each cartridge were clustered and annotated separately and used as a control for the batch effect correction validation. In total, six methods were evaluated for mRNA batch effect correction: 1) Canonical Correlation Analysis (CCA), 2) Batch balanced K-Nearest Neighbors (BBKNN), 3) Harmony, 4) Scanorama, 5) scVI and 6) totalVI. After extensive evaluation, the CCA method was selected for downstream analyses using Seurat R package version 4.1.1. The unified merged object was split into individual samples.

Scaling and normalization were performed, log normalization for mRNA counts and the centered log-ratio transformed (CLR) for AbSeq. All the mRNA features common between the analyzed samples were considered as variables and were used for the identification of integration anchors between the individual datasets. For picking anchors, the FindIntegrationAnchors function was used. CCA dimensional reduction was applied and first 30 dimensions were used to specify a neighbor search space with a final selection of 20 neighbors. Dataset integration was performed using the IntegrateData function, employing the first 30 CCA dimensions. The corrected feature count matrix was stored as an individual assay in the merged Seurat object.

## Clustering analysis

The merged unified dataset was processed for a cell clustering analysis. The analysis was performed for each modality separately. For the mRNA assay, read counts were normalized at the cell level using the "LogNormalize" method. Then they were scaled and centered at the gene level by applying the scaleData function on the normalized count matrix. The transformed count matrix was then used as an input for applying linear dimensionality reduction; Principal Component Analysis (PCA) using the RunPCA function.

To perform graph-based community detection, initially, a shared nearest-neighbor (SNN) graph was constructed using the 20 first Principal Components (PCs) for the merged dataset using the FindNeighbors function. The SNN graph was further processed by using the FindClusters function to determine the cell clusters with a resolution of 0.8 (Louvain algorithm-based clustering). To visually inspect the heterogeneity of the datasets through cell clusters, uniform manifold approximation and projection (UMAP) non-linear dimensionality reduction was performed by applying the RunUMAP function on the same PCs that were used for clustering. Visualizations were generated in R using the ggplot2 (v. 3.4.0 or later), and RColorBrewer (v. 1.1.3 or later).

## Differential expression analysis

Log-normalized RNA and centered log ratio (CLR) normalized AbSeq counts were used for differential expression analysis and visualization. The list of cluster gene markers was determined by using the FindAllMarkers function (Wilcoxon Rank Sum test) with a minimum percent of cells expressing set up to 0.20 and log$_2$ fold change >1 or 0.25, for CD34$^+$ HSPCs and re-clustered immature HSPCs, respectively. The same function was used for calculating differentially expressed proteins in HSC-1 vs HSC-2, with minimum percent of cells expressing set up to 0.25 and no fold change filter. Differentially expressed genes and proteins were visualized by FeaturePlot expression patterns across the UMAPs, dotplots and violin plots using the ggplot2 (v. 3.4.0), and RColorBrewer (v. 1.1.3).

For the pseudobulk analyses of immature subsets, we used unnormalized count matrix, and pseudobulks were generated per sample for each relevant cluster with the function pseudobulk() from the R package glmGamPois v.1.12.2. Differential gene expression analysis taking these pseudobulks as input was then performed with the DESeq2 R package v1.40.2 using the function DESeq(). When comparing clusters HSC-1 vs HSC-2, the sample of origin was considered as a covariate in the design formula ( ~ sample+condition) (adjusted $p$-value threshold <0.05 for differentially expressed genes).

## Cell label transfer

All clusters were initially manually annotated using previously published papers and dataset, mostly relying on well-known marker genes

and cell surface proteins. To confirm our annotations, we utilized existing WTA dataset of full bone marrow cells published by Triana et al.[8]. The reference dataset was log normalized for mRNA counts, scaled using the ScaleData function, followed by reduction PCA by employing RunPCA function.

FindTransferAnchors function was used to find anchors for each Seurat object (k = 30), where "Prediction_HCA" labels were used to transfer names to the query dataset using TransferData function. The predictions were added to metadata under the name "Predicted_Label." There was no cut off value selected.

## Immature compartment analysis
To construct cell lineages from the single cell targeted gene expression data, data were subset to include HSC/MPP, MPP/MK-Ery, MPP/LMPP, MEP, MDP-1, GMP-1, GMP-Neut, LyP. Principal components (PCs) were re-calculated and cells were re-clustered with 17 PCs and resolution 0.4, which yielded 9 clusters. The embedding was used as input for trajectory analysis by Slingshot (2.4.0). Cluster 0, the most immature HSC-1 cluster, was set as the starting cluster. Genes changing expression throughout the trajectories were calculated with tradeSeq (1.10.0)[43] and visualized using "patternTest", by assessing the differences in expression patterns between lineages. First, comparison between all pairs (6 pairs) was performed defining the significance of the adjusted $p$-value (FDA) < 0.05. Then, comparisons were compiled and genes that were picked up as significant in any of the comparisons were plotted (Supplementary Data 13).

## FACS sorting for molecular and functional experiments
G-CSF-mobilized apheresis samples from healthy donors were processed within 24 h. MNCs were isolated by density gradient centrifugation using Pancoll (PAN Biotech). CD34 enrichment was performed using the CD34 MicroBead Kit (Miltenyi Biotec) according to the manufacturer's protocol. For flow-cytometry based analysis and sorting of marker-defined HSPC subpopulations, lineage-committed cells were stained with a cocktail of biotinylated lineage-specific antibodies (CD2, CD3, CD14, CD16, CD19, CD56, CD235a) and Streptavidin-PE-Dazzle. To gate for viable HSPC subpopulations, MNCs were stained with fluorophore-labelled antibodies binding CD34 (APC), CD38 (APC-H7), CD45RA (BV570), CD90 (PerCP-Cy5.5) and CD273 (BV711) and Fixable Viability Dye (EF506, Invitrogen) (Supplementary Data 17). FACS was performed on a BD FACSAria III instrument (BD Biosciences).

## In vitro expansion and differentiation of primary HSPCs
FACS-sorted lin−CD34+CD38−CD45RA−CD273high and lin−CD34+CD38−CD45RA−CD273low HSPCs were counted, and 3000 cells were seeded in 200 µl SFEM II (StemCell Technologies) supplemented with the human cytokines (all from Peprotech) 100 ng/ml SCF, 100 ng/ml TPO, 100 ng/ml FLT-3, 30 ng/ml IL3, and supplements 38 nM UM171 and 4ug/mL human-LDL (both from Stem Cell Technologies). Cells were cultured at 37 °C, 5% $CO_2$ for 7 days. All individual conditions were plated as duplicates. On the day of analysis, the cells were harvested and 10 µl of cell suspension was used for cell counting. The remaining cells were spun down (290 x g, 7 min, 4 °C) and resuspend in 50 µl Brilliant Stain Buffer (BD Biosciences). Cells were stained with fluorochrome-labeled antibodies against CD201 (EPCR), CD90, CD34, CD45RA, CD133 and Fixable Viability Dye to determine the differentiation status by flow cytometry using a BD LSRFortessa (BD Biosciences).

## Time-lapse imaging and single cell tracking
Cells were imaged using video-microscopy and tracked as previously described[57,58]. Fifty FACS-sorted lin−CD34+CD38−CD45RA−CD273high and lin-CD34+CD38−CD45RA−CD273low HSPCs were seeded in a 24-well NUNC tissue culture plate (Thermo Fisher Scientific) equipped with 4 well micro-inserts (IBIDI). Plates were gas-tight sealed with adhesive

tape after 5% $CO_2$ saturation and placed inside a CellObserver Z1 microscope (Zeiss) equipped with a 37 °C temperature module to maintain cell culture. Phase contrast images were acquired every 2 min using a 10x phase contrast objective, and an AxioCamHRm camera (at 1388 × 1040 pixel resolution) with a self-written VBA module remote controlling Zeiss AxioVision 4.8 software. Cells were imaged for 7 days and manually tracked using a self-written computer program (The Tracking Tool TTT) developed by Timm Schroeder[57] until the fate of all progenies in the third cell generation was determined. Time of first division was defined as the time from the start of the time-lapse recordings to the first cell division (cytokinesis). Dead cells were depicted by their shrunk, non-refracting appearance with immobility. The current analysis did not rely on data generated by an unsupervised computer algorithm for automated tracking.

## Protein analysis
5000 lin−CD34+CD38−CD45RA−CD273high and lin−CD34+CD38−CD45RA−CD273low HSPCs were FACS-sorted into 15 µl lysis buffer on ice (M-PER™ Mammalian Protein Extraction Reagent (Thermo Fisher Scientific, USA) supplemented with Pierce Protease and Phosphatase Inhibitor Mini Tablets (Thermo Fisher Scientific). 3 µl of cell lysate per sample and lane was used for protein detection. Cartridge preparation, sample processing and applying on the capillary was performed according to the manufacturer's protocol of the 12–230 kDa Jess or Wes Separation Module, 8 × 25 capillary cartridges kit (Protein Simple). All primary antibodies were diluted in ratio 1:30. Protein detection was validated using High-Dynamic-Range (HDR) multi-image analysis for every tested sample and analysis was performed with the software Compass SW V5.0.0 (Protein Simple).

## RNA-Seq of sorted populations
RNA was isolated from 1 - 2 ×10⁴ FACS-sorted lin-CD34+CD38-CD273high and lin-CD34+CD38-CD273low HSPCs using the miRNeasy micro kit (Qiagen). RNA quality was validated by using high sensitivity RNA screen tape analysis utilizing the 4200 TapeStation System (Agilent). Approximately 1–5 ng of total RNA was used as starting material for SMART®-Seq HT Kit (Takara). Sequencing was performed on the NextSeq500 sequencer (Illumina) using v2 chemistry with 1 × 75 bp single-end setup.

## Colony-forming unit assay
Primary colony-forming unit assays (CFUs) were performed by seeding 300 FACS-sorted cells from the following populations: lin−CD34+CD38−CD45RA−, lin−CD34+CD38−CD45RA−CD273high, lin−CD34+CD38−CD45RA−CD273low and lin−CD34+CD38−CD45RA−CD90+CD49f+ in 3 ml of MethoCult™ medium (H4034, STEMCELL Technologies). From this, 1 ml—equivalent to approximately 100 cells per plate—was plated into each 35 mm dish. Plates were incubated at 37 °C with 5% $CO_2$ for 14 days, after which colony formation and lineage distribution were assessed microscopically (CellObserver, Zeiss). For the secondary assay, cells were harvested from CFUs and counted. For each sample, 30,000 cells were resuspended in 3 ml MethoCult™ and divided into plates at 10,000 cells per plate. Secondary colonies were then scored after an additional 14 days of incubation at 37 °C with 5% $CO_2$.

## Mixed lymphocyte reaction assay
Mixed lymphocyte reaction (MLR) assays were performed by allogeneic co-culturing of 1 × 10⁴ CD34+ HSPCs with 1 × 10⁴ CD3+ T-cells. Both cell-types were MACS-enriched from mobilized apheresates of healthy donors. Prior to co-culture, T-cells were activated using anti-CD2/3/28 coated beads (Miltenyi Biotec) for 2 h. HSPCs were incubated for 2 h with 20 µg of either neutralizing anti-PD-L2 antibody or goat IgG isotype control (both R&D Systems). Anti-PD-L2 and IgG co-cultures were performed in comparison to mono-culture of pre-activated T-cells in 96-well plate format for 72 h at 37 °C in 5% $CO_2$ and a relative

humidity of 95%. HSPCs and T-cells before and during co-cultures, activation, or antibody incubation were cultivated in SFEM II (Stem Cell Technologies), supplemented with 100 ng/ml human SCF (Peprotech), 100 ng/ml human TPO (Peprotech), 100 ng/ml human FLT-3L (Peprotech), 30 ng/ml human IL-3 (Peprotech), 50 U/mL human IL-2 (Peprotech), 38 nM UM171 (Peprotech), 50 ng/ml human LDL (Peprotech) and 1% Penicillin/Streptomycin (Thermo Fisher). After 72 h co-culture, the below described assays were performed with cells, and supernatants were snap-frozen and stored below −70 °C until cytokine profiling.

## T-cell proliferation and activation assay

T-cell proliferation was monitored before and after MLR using Cell-Trace™ Violet (Thermo Fisher Scientific) staining and fluorescence assessment according to the manufacturer's protocol. Proliferation index was calculated as previously described[50]. $CD8^+$ T-cell activation upon MLR was monitored using flow-cytometric quantification of the $CD69^{hi}$ fraction of $CD3^+CD8^+$ T-cells. CD69 is a surrogate marker for $CD8^+$ T-cell activation[45] and was verified by comparison of $CD69^{hi}$ fractions of $CD3^+CD8^+$ T-cells before and after bead-based activation. The staining panel comprised CellTrace Violet, CD3-PerCP-Cy5.5 (clone OKT3, BD), CD4-FITC (clone SK3, BD), CD8-APC-H7 (clone HIT8a, BD), CD25-PE (clone 2AE, BD), CD34-APC (clone 8G12, BD), CD69-R718 (clone FN50), CD273-BV711 (clone MIH18, BD). Cells of interest were gated based on SSC and FSC to exclude debris, and singlets subgated with proportional FSC-A to FSC-W signals. From singlets, $CD3^+CD34^-$ T-cells were gated for $CD4^+CD8^-$ and $CD8^+CD4^-$ T-cells (Supplementary Figs. S11, S14). All gates were set using FMO controls. Samples were acquired and analyzed on a BD LSRFortessa or BD FACSCelesta flow-cytometer using BD FACSDiva Software v8.0.1 and FlowJo 10.9.0. (BD).

## Memory and regulatory T-cell phenotype assessment

Memory phenotypes of $CD3^+CD4^+$ T-cells were assessed after MLR using CCR7 and CD45RA surface staining and flow-cytometric quantification. The $CD25^{hi}FOXP3^+$ $T_{reg}$ fraction of $CD3^+CD4^+$ T-cells was quantified upon MLR using surface and intracellular staining followed by flow-cytometric measurement. Intracellular staining was performed using the BD Pharmingen™ Transcription Factor Buffer Set according to the manufacturer's protocol.

The T-cell differentiation panel comprised CD3-PerCP-Cy5.5 (clone OKT3, BD), CD4-FITC (clone SK3, BD), CD8-APC-H7 (clone HIT8a, BD), CD25-PE (clone 2AE, BD), CD34-APC (clone 8G12, BD), CD45RA-BV570 (clone HI100, BD), CD197-RB780 (clone 2-L1-A, BD), FOXP3-R718 (clone 236 A/E7, BD). $CD3^+CD4^+CD8^-$ and $CD3^+CD4^-CD8^+$ T-cells were gated as in the T-cell proliferation and activation panel. Memory T-cell populations were gated from $CD3^+CD4^+CD8^-$ T-cells using quadrant gates based on CCR7 (CD197) and CD45RA signals: Naïve T-cells ($T_N$): $CCR7^+CD45RA^+$; central memory T-cells ($T_{CM}$): $CCR7^+CD45RA^-$; effector memory T-cells ($T_{EM}$): $CCR7^-CD45RA^-$; terminally differentiated effector memory T-cells re-expressing CD45RA ($T_{EMRA}$): $CCR7^-CD45RA^+$. The $CD25^{hi}FOXP3^+$ population was gated from $CD3^+CD4^+CD8^-$ T-cells to quantify $T_{regs}$. All gates were set using FMO controls. Samples were acquired and analyzed on a BD LSRFortessa or BD FACSCelesta flow-cytometer using BD FACSDiva Software v8.0.1 and FlowJo 10.9.0. (BD).

## Cytokine profiling

Frozen MLR supernatants were thawed and the levels of IFN-γ, TNF-α, IL-6, IL-17A, Granzyme A, Granzyme B, Perforin and Granulysine quantified using the LEGENDplex cytokine bead array assay (Biolegend) on a BD FACSCelesta™ Analyzer system with a high-throughput sampler according to the manufacturer's recommendations.

## Statistics

Statistical analyses were performed in GraphPad Prism (v. 9.5.1) and Rstudio. All sample data was checked for normality of distribution to determine the accurate test. Applied tests and statistics are indicated in the figure legends.

## Reporting summary

Further information on research design is available in the Nature Portfolio Reporting Summary linked to this article.

## Data availability

The Transcriptomic/AbSeq expression data generated in this study is available in the ArrayExpress repository under accession code E-MTAB-14596. Source data are provided with this paper.

## Code availability

R code is available at: https://github.com/TessaSchm/Early-HSPC-differentiation/tree/main.

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

## Acknowledgements

We would like to thank all study participants and healthcare professionals involved in sample collection, especially Ola Rolfson from the Department of Orthopaedics at the University of Gothenburg. Additionally, we are very grateful to Edyta Kowalczyk and Yujuan Gui from BD Biosciences for their immense work with initial bioinformatic analyses and collaboration throughout the years, as well as the other members of

the BD Multiomic Alliance. This study was supported by grants from the Deutsche Jose Carreras Leukämie-Stiftung (DJCLS 11 R/2020 and DJCLS 15 R/2023), the Deutsche Forschungsgemeinschaft DFG (RI 2462/9-1 and RI 2462/10-1), and the hessian LOEWE Funding Program (Hessen State Ministry for Higher Education, Research and the Arts, III L 5 – 519/03/03.001 – [0015] and III 5.7 - 519/03/10.001-(0004)). Further support was provided by the Centro de Investigación Biomédica en Red – Área de Oncología - del Instituto de Salud Carlos III (CIBERONC; CB16/12/00369 and CB16/12/00489), Instituto de Salud Carlos III/Subdirección General de Investigación Sanitaria (FIS No. PI16/01661 and PI23/01331), the Swedish Research Council (2020-02783), the Swedish Cancer Society (22-2388), the Swedish state via the ALF agreement (ALFGBG-963642), the Assar Gabrielsson Foundation, the Wilhelm and Martina Lundgren Research Foundation, and by the Cancer Research UK [C355/A26819] FCAECC and AIRC under the Accelerator Award Program (EDITOR). The authors acknowledge support from the National Genomics Infrastructure in Stockholm funded by Science for Life Laboratory, the Knut and Alice Wallenberg Foundation and the Swedish Research Council, and SNIC/Uppsala Multidisciplinary Center for Advanced Computational Science for assistance with massively parallel sequencing and access to the UPPMAX computational infrastructure. We further thank Stefan Guenther from the Genomics Core Facility of the Max-Planck Institute Bad Nauheim, Germany, for his support in next generation sequencing. Some figure panels have been created with BioRender.com.

## Author contributions

H.K., T.S., C.S and M.K. contributed equally to this work. H.K., M.S.N., C.S. and T.S collected and processed bone marrow and blood samples with the help of F.P. H.K., M.S.N., C.S., T.S., W.Y., and M.K. planned and performed the experiments, under the supervision of F.B.T., B.P., H.B. and M.A.R. H.K., T.S., C.S., W.Y. and C.G. performed bioinformatic analyses, A.J. supervised the bioinformatic analyses. H.K., T.S., C.S., M.K., B.P., F.B.T. and M.A.R. wrote the manuscript. All authors read and approved the manuscript.

## Funding

## Competing interests

The authors received research funding in the form of free reagents from BD Biosciences within the BD Multiomics Alliance. H.B. has received licensing fees and royalties from Medac, research support from Erydel, Miltenyi, Sandoz-Hexal (a Novartis company), honoraria or speaker fees from Medac, Miltenyi, Novartis and Terumo BCT, consultancy or membership in advisory boards for Apriligen, Arensia, Boehringer-Ingelheim Vetmed, Celgene (a BMS company), Editas, Medac, NMDP, Novartis, Provirex and Sandoz-Hexal, and acknowledges stock ownership in Healthineers, none of which are of relevance to the work at hand. B.P. reports honoraria for lectures from and membership on advisory boards with Adaptive, Amgen, Becton Dickinson, Bristol-Myers Squibb-Celgene, Janssen, Merck, Novartis, Roche, Sanofi and Takeda; unrestricted grants from Bristol-Myers Squibb-Celgene, EngMab, Roche, Sanofi, and Takeda; and consultancy for Bristol-Myers Squibb-Celgene, Janssen, Sanofi, and Takeda, none of which are of relevance to the work at hand. The remaining authors declare no competing interests.
