## [Peer Review file · Nature Communications]

Continuous map of early hematopoietic stem cell differentiation across human lifetime

Corresponding Author: Professor Michael Rieger

Version 0:

Reviewer comments:

Reviewer #1

(Remarks to the Author)

Komic et al. used Abseq, a single-cell multi-omics tool that here measures 596 genes and 46 surface marker expression to profile over 62,000 CD34+ HSPCs from 15 donors spanning young, mid-age and old. The authors aligned their data to Triana et al annotations (Abseq) and carefully explained top marker genes and differential surface markers. They further performed pseudotime and trajectory analysis to hypothesize young donor's HSC is more productive. Interestingly, the authors checked differential markers among more primitive HSCs and found CD273 can be used to enrich for cells of better HSC capability. Some of the observations are interesting, yet the reviewer has several concerns.

First, there are already two papers in Nature Immunology characterizing bone marrow stem and progenitors (RNA+Surface marker) at much larger scale, Triana et al 2021 (Abseq, 462 genes, 97-197 surface markers, 122,004 cells) and Zhang et al 2024 (CITE-seq+ Matching Flow, whole transcriptome, 132-266 surface markers, >300,000 CD34+ cells), the authors should emphasize the additional insights either technical or biological that this dataset could provide in addition to studying/analyzing the existing two. For example, many marker genes and surface markers have been reported previously, are they consistently found in this dataset. The reviewer thinks the authors should delve deeper into the function of interesting findings such as CD273 and its interaction in niche (it's a PD ligand) by KO or antibody blocking and build a story about it.

Line 474, the author claim they found several new candidate markers for some cell states, please make sure to check their expression in published bone marrow atlas. I'm pretty sure HLF has been reported numerous times.

The authors need to change CITE-seq to Abseq to avoid confusion. Cellular Indexing of Transcriptomes and Epitopes by Sequencing (CITE-seq) is a multimodal single cell phenotyping method developed in the Technology Innovation lab at the New York Genome Center. CITE-seq uses 10X 3' or 5' kit (whole transcriptome, 10X genomics) and ADTs (Biolegend) where Abseq the authors used is a targeted transcriptome plus antibody method developed by BD Rhapsody (well-based). Check Simon Hass lab 2021 Nature Immunology paper for reference. Also, it should be tradeSeq instead of TRADE-seq, make sure to get all these nomenclatures right.

Secondly, pseudotime, trajectory analysis and to larger extend, scRNA-seq provides hypothesis and should be carefully examined and validated by functional readout. A stem cell is define based on its self-renewal and multipotency not by just a few gene expression. Hypothesis such as difference of cell differentiation or fate in this study need to be validated by flow cytometry (accumulation of certain immunophenotypic defined progenitor) and colony formation assays to examine lineage output (CFU). Specially, CD273+ as a stem cell marker, can CD34+CD273+ cells self-renew and give rise all lineages like CD34+CD38-CD90+CD45RA- HSC?

The author cultured sorted CD273hi cells for 7 days in SFEM II with cytokines and then were flow profiled for surface markers, checked for gene expression and cycle entry delay. These are good clues but insufficient to profile a stem cell without functional readout such as long-term culture, CFU to exam lineage output, and xenograft.

Finally, data transparency. Data and code are not available, and many statistical/biological details and supplementary data

are missing, making it impossible to judge the rigor of this study.

Data availability is not in line with Nature journals joint policies. The authors should provide reviewer link to the sequencing data repository instead of asking the editor to request access. Also, all code should be readily available on public platforms eg. github at the time of review instead of "available upon a scientifically reasonable request". Flow data is recommended to be uploaded to flowrepository.

Current supplementary information is not adequate nor transparent that the reviewer or potential readers could examine the data or reproduce their findings. For instances, author should provide a supplementary table to list all samples by age, sex and race, and also the cell frequencies in each. A supplementary table to list all cell barcode to cell annotations and correlation score. Supplementary tables to list differentially regulated surface marker expression. Authors should also explain what cutoff is used to call doublets, cell filtering and assigning cells to each sample tag, and provide processed filtered matrix as raw data (not possible to see if they are deposited without access) or excel sheet. Furthermore, the statistical significance or parameters are missing, for example How was label transfer performed, based on what method and cutoff?

The authors should explain by what statistical standard did they use to select "596 highly relevant genes for human hematopoiesis". To defend such selection, the authors should rigorously examine their data complexity and cell annotations with full single-cell transcriptome datasets to make sure the targeted transcriptome is equivalent and cell composition is comparable. For instance, there is a more comprehensive (>300k single cells) CITE-seq atlas of human bone marrow CD34+ progenitors in a Nature Immunology paper by Lee Grimes lab earlier this year.

Line 505, mobilized CD34+ cells and bone marrow cells could be very different and should not be used to confirm findings from bone marrow (niche).

Line 509, it is unclear what flow cytometry gate was used and how was it determined for sorting CD273 high and CD273 low cells (assuming both express CD273?). The number of cells in each gate is critical but missing. Judging by the flow data in Figure S9, CD273 positive cells seems very few (0.1%?) which makes me wonder if the cell frequency you see in flow match what's in the Abseq. This information is very important to evaluate whether CD273 can be used as good flow marker.

Minor concerns,

Line 275, citation errors

Line 323, "Forty-six oligo nucleotide-labeled antibodies commonly used to characterize HSPCs and hematopoietic cells by flow cytometry were included to capture the surface proteome of the same cells." Add rationale or citation for "commonly".

Line 365, remove "comprehensive" here and its appearance in most cases. 60,000 cells of human CD34+ is very small compared to studies of recent years and most immature stem cells recognized as human "HSC" (CD34+CD38-CD90+CD45RA-) is only 2% of CD34+ cells (~1200).

Figure 6, need to specify cells from which donor is used for each panel, are they donor matched?

(Remarks on code availability)
code is not readily available to the reviewer.

Reviewer #2

(Remarks to the Author)

This manuscript systematically investigates the early gene network changes in human hematopoietic stem cells (HSCs) during differentiation using a targeted single-cell CITE-seq approach on FACS-enriched bone marrow stem and progenitor cells from 15 healthy donors. Their analyses show young donors exhibited more productive differentiation from HSCs to committed progenitors. Conversely, the study reveals a delayed differentiation to all lineages in the elderly, with a pronounced effect in the lymphoid lineage. This finding could help explain the overabundance of HSCs in the elderly, accompanied by a myeloid shift in further differentiated cells. TRADE-Seq analysis showed continuous gene expression changes, highlighting both known HSC-related genes and new immature marker genes (DLK1, ADGRG6), providing a gene expression roadmap at the earliest branching points. The study also identified CD273 (PD-L2) as highly expressed in the most immature stem cells. The study demonstrates strength in the accurate quantification of lowly expressed genes. This is achieved through the targeted approach of deep scCITE-seq, which comes with a low dropout rate for the targeted genes. This paper is overall well written, although more structuring of the discussion section (which has a very long initial paragraph) would be helpful.

The manuscript lacks clarity regarding the motivation for assessing the expression of genes associated with clonal haematopoiesis (CH). Furthermore, the interpretation of the outcomes related to these findings is not adequately addressed, resulting in a disjointed paragraph that appears without sufficient context. Were any CH mutations found in the scCITE-seq data? Or would these be 'invisible' to the targeted approach taken?

While DLK1 and ADGRG6 could be relevant immature marker genes further functional validations are necessary to better understand the roles of these markers in HSC biology.

Line 441: The text states that there was no reduction of LYP, which does not appear to be in line with the remainder of their results presented. Clarification is therefore needed: did the authors find a reduction or not?

Line 400: The identification of new immature marker genes such as ADGRG6 raises the question whether the authors can find corroborating evidence from other datasets, including published expression data?

Data and Code Availability: While I did see the data generated in a zenodo repository, it is crucial for the CITE-seq data to be published in an international repository prior to the acceptance of the manuscript. Similarly, could the authors clarify whether they have decided to release the analysis code openly, or will it remain restricted? I do see a github repo, yet the paper suggests code may not be made freely available? Restricting code would require further justification, as it significantly impacts the reproducibility and transparency of the study.

(Remarks on code availability)

Version 1:

Reviewer comments:

Reviewer #1

(Remarks to the Author)

The reviewer has some questions (see attached figure) about the now provided gating illustration of sorted CD273+ cells in Figure S9. There is a consensus on the immunophenotype of human HSCs: Lin-CD34+CD38-CD45RA-CD90+ and MPPs: Lin-CD34+CD38-CD45RA-CD90-. From the new S9a (now there are more cells), it is clear that the majority of CD273+ cells in BM or PB are CD90-, which falls more into the MPP immunophenotype. The authors did not answer how CD273-high and low cells were gated for sorting (Q1), so I have to just assume it's based on the line I draw. I'm also confused how CD34+CD38-CD45RA-CD90+ HSCs could have a higher MFI than CD34+CD38-CD45RA- HSC&MPP because there are only 1–2 CD273-high HSCs (arrow pointed) in the flow (Q2). CD49f+, CD133+, EPCR+ gating (Fig. 6B) are missing, so make sure to provide representative flow plots whenever a gate is used. The cell frequency/percentage of each gate should also be clearly labeled on the plots. Make sure to present flow data in a clear and transparent way because it is crucial to the major innovation of this paper. The authors used the two terms HSCs (by their definition, CD34+CD38-CD45RA-Lin-) and HSPCs inconsistently in this paragraph, which is very confusing and should all be corrected to HSPCs. The flow illustration (with proper labeling) should be moved to the main figure because it is important to know whether this proposed marker can be used to sort enough cells (if readers can use it for their experiments). The subtitle of that paragraph might need to change because there are very few CD273+ cells in the most immature HSCs (CD90+), and validation was done on early HSPCs (mostly CD45RA- MPPs). These sorted CD273+ cells are a mixture of MPPs (90%) and HSCs (10%), are quiescent, and have the same lineage output. I think it is compelling that CD273 is a marker of quiescence among these early HSPCs; however, "most immature" HSPCs are stem cells, and there is no evidence of what impact CD273 has on CD90+ HSCs. Please avoid making claims regarding "most immature" or "stem cell-like" in lines 52–53 to avoid confusion.

Regarding the new data in Figure 7, I'm not sure if it's appropriate to use CD34+ HSPCs to prove CD273 protects immature HSPCs. CD34+CD38- is about 10–15% of CD34+ cells, CD34+CD38-CD45RA- is about 60% of CD34+CD38-, and CD34+CD38-CD45RA-CD273+ cells are 5% of CD34+CD38-CD45RA-. Therefore, out of the 10,000 CD34+ in co-cultures, there are probably just ~30 CD34+CD38-CD45RA-CD273+ cells. The data showed CD273 antibody blockade restored T cell activation in co-culture, and I found it hard to believe just 30 CD34+CD38-CD45RA-CD273+ cells could inhibit T cell proliferation. In other words, are there other CD273+ cells within CD34+ compartments? This could easily be examined by comparing the number of CD34+CD273+ to CD34+CD38-CD45RA-CD273+ cells in the flow. Or the authors could sort CD34+CD38-CD45RA-CD273+/high cells vs. CD34+CD38-CD45RA-CD273-/low cells and test their immunomodulatory effect on immune cells in co-culture.

The authors have shown advantages of their approach and dataset over the two existing atlases respectively: 1) More early CD34+ HSPCs than Triana/Haas'; 2) Targeted Ab-seq provides a better profiling of low-expression genes such as MPL compared to Zhang/Grimes'. It would be useful to provide a list of these "missed" genes, including the ones provided in the rebuttal letter, in the supplementary table and mention them in the discussion.

Use Abseq, not CITE-seq. BD already trademarked Abseq to make it their own thing to compete against CITE-seq and Biogend Totalseq. BD would much prefer you use Abseq instead of CITE-seq. As for "searchable," Triana/Haas used Abseq consistently (never mentioned CITE-seq) in their 2016 Nature Immunology paper, and it has 54k accesses and an altmetric score of 245.

The added CD273 function data/research direction has now better strengthened the biological findings, making it the highlight of this study. CD273 is a new immunomodulatory marker that helps early HSPCs. Since there is no major innovation in analyzing the data, the reviewer suggests limiting the paragraph and figures describing markers and unvalidated hypotheses (transcriptome clusters/trajectories without functional isolation and validation) that are just consistent with the literature. Some suggestions as below.

The revised abstract, line 49: change “demonstrate” to “suggest.” Trajectory and pseudotime analysis without functional validation (e.g., cell tag tracing or captures at different time points or index sort) do not have the ability to demonstrate or prove anything. Line 55: remove “comprehensive” per discussion before.

Notta/John Dick already have that concept of Mk/E branching from early HSC/MPP in their 2016 Science paper. The “CITE-seq of CD34+ HSPCs reveals early Mk/E lineage branching” paragraph (as the first and main claim) provides nothing conceptually new but more of a technical description of the dataset, and most of it can be moved to the method. This whole paragraph simply shows manually annotated HSC/MPP are transcriptomic stem and progenitors.

Myeloid bias and increased HSPCs in old people are well-known.

Figure 4 provides little readable information and can be moved to the supplemental.

The discussion is too long; just focus on biological findings and avoid too much speculation. Lines 385–396: Although I agree and believe hematopoiesis is a stepwise commitment rather than a “cloud,” it takes a lot more to claim these discrete cell states, such as by isolating transcriptomic cell populations and functionally validating them.

(Remarks on code availability)

Data and code transparency is greatly improved.

Reviewer #2

(Remarks to the Author)

The authors have carefully responded to the reviewer questions. I am happy with the revised version.

(Remarks on code availability)

Version 2:

Reviewer comments:

Reviewer #1

(Remarks to the Author)

The authors have addressed my concerns.

(Remarks on code availability)

Rebuttal letter with point-by-point replies to the reviewers' questions

We would like to express our deepest gratitude to both reviewers and the editorial team for taking the time to review our study, and to provide constructive and very helpful comments to improve our manuscript. We acknowledge their positive feedback to our study and their detailed descriptions of aspects to be improved, which were addressed in the revised version of the manuscript. After performing additional experiments and a thorough analysis of previously published single cell sequencing data, we added new main figures, supplementary figures and tables. As recommended by Reviewer #1, we extended our study in investigating the immunomodulatory function of the described surface marker CD273/PD-L2 on immature HSCs in their interplay with T-cell activation, proliferation, differentiation and cytokine-release (new Figure 7).

Point-by-Point answers to the reviewer comments

Reviewer #1 (Remarks to the Author):

Komic et al. used Abseq, a single-cell multi-omics tool that here measures 596 genes and 46 surface marker expression to profile over 62,000 CD34+ HSPCs from 15 donors spanning young, mid-age and old. The authors aligned their data to Triana et al annotations (Abseq) and carefully explained top marker genes and differential surface makers. They further performed pseudotime and trajectory analysis to hypothesize young donor's HSC is more productive. Interestingly, the authors checked differential markers among more primitive HSCs and found CD273 can be used to enrich for cells of better HSC capability. Some of the observations are interesting, yet the reviewer has several concerns.

First, there are already two papers in Nature Immunology characterizing bone marrow stem and progenitors (RNA+Surface marker) at much larger scale, Triana et al 2021 (Abseq, 462 genes, 97-197 surface markers, 122,004 cells) and Zhang et al 2024 (CITE-seq+ Matching Flow, whole transcriptome, 132-266 surface markers, >300,000 CD34+ cells), the authors should emphasize the additional insights either technical or biological that this dataset could provide in addition to studying/analyzing the existing two. For example, many marker genes and surface markers have been reported previously, are they consistently found in this dataset. The reviewer thinks the authors should delve deeper into the function of interesting findings such as CD273 and its interaction in niche (it's a PD ligand) by KO or antibody blocking and build a story about it.

#Reply 1: Thank you for raising these important points and suggestions. To address the Reviewer's concerns, we followed the recommendations to carefully analyze the previously published studies and data sets of Triana *et al.* and Zhang *et al.*, to allow direct comparison of our findings and to emphasize the strengths and specific contributions of our dataset, as highlighted below.

Triana, S. *et al.* *Nature immunology* **22**, 1577–1589; 10.1038/s41590-021-01059-0 (2021).

Zhang, X. *et al.* *Nature immunology* **25**, 703–715; 10.1038/s41590-024-01782-4 (2024).

Number of cells for single cell profiling

We carefully investigated the data published by Triana *et al.* and Zhang *et al.* and reconstructed the workflow of these studies. While both studies used an impressive number of cells, the number of CD34+ BM cells included in their studies was approximately in between 10,000 and 30,000 (*see table below*). In our study, we analyzed 62,277 CD34+ BM cells. Furthermore, we used 15 donors covering three age groups, which was a considerable higher number of donors than in these published studies.

	Komic et al. Targeted + 46 AbSeqs	Triana et al. WTA + 97 AbSeqs	Triana et al. targeted + 97 AbSeqs	Zhang et al. WTA + 132 AbSeqs
# Donors	n = 15	n = 1	n = 6	n = 4
CD34+ cells	62 277	13 165	10 361	28 549

The workflow of both studies including cell numbers and donors are shown here (Letter Fig. 1):

Letter Fig. 1. Overview of analyzed bone marrow cells in previously published studies. Flow charts depicting the number of cells included in each part of Triana *et al.* (A) and Zhang *et al.* (B). (C) Distribution of CD34+ and CD34- cells per donor (n=6) used in the study by Triana *et al.* (D) Total amount of cells and distribution of CD34+ cells in the study by Triana *et al.*

Comparison of gene expression in immature HSPC clusters with published data sets

- 1) The selection of genes in our panel enabled us to reconstruct early human hematopoietic differentiation. Thanks to the studies of Triana *et al.* and Zhang *et al.*, cell label transfer and cell/cluster-type correlation analyses demonstrated a high concordance of our cluster and cell annotation with the clusters described by whole transcriptome analyses (WTA)-based data from these comprehensive studies (shown in new Suppl. Figure S5. and new Suppl. Table 5). *See more details and figures in #Reply 7.*
- 2) Gene markers of the most immature cell cluster identified in the two published studies showed a high overlap (new Suppl. Table 8) with our gene marker list of the HSC-1 cluster confirming that the studies using different sequencing approaches enriched for similar populations.
- 3) When applying the gene score of our reported HSC-1 cluster to the WTA dataset from Zhang *et al.*, we found the highest enrichment in their most immature cluster HSC1. See Letter Figure 2.

Letter Fig. 2. Correlation of clusters from Zhang *et al.* with our cluster HSC-1 score. The clusters with the highest correlation were ordered as such: HSC-1, HSC-2, MPP-1, MPP-2, LMPP-1-cycling, ...

- 4) We next compared differentially expressed genes between our HSC-1 and HSC-2 clusters with gene expression values received from the Triana *et al.* and Zhang *et al.* datasets.

Triana *et al.* used a targeted gene panel to reconstruct the whole hematopoietic hierarchy. While we rationally designed our gene panel and our CITE-seq approach to resolve early events in human HSC differentiation and lineage choice, they focused on combining the best marker genes for all hematopoietic cell types. The immature HSPC fraction analyzed by Triana *et al.* is naturally small in relation to all analyzed BM cells and they nominated the most immature cluster as “HSC/MPP”. Shared genes up-regulated in the HSC/MPP cluster and in our HSC-1 cluster include *MPL*, *CD34*, *CRHBP* and *HOPX* (see Letter Fig. 3). Primers for *MMRNI*, *MLLT3*, *ADGRG6* and *HLA-E* were not included in their targeted panel, which we found significantly upregulated in HSC-1, among others.

In the WTA dataset from Triana *et al.* (from 1 donor), increased expression of *ADGRG6*, *HLA-E* and *MMRNI* was observed in HSC/MPP cluster cells as compared to more lineage-restricted progenitors, which is consistent with our results of increased expression of these genes in HSC-1 cluster cells. By contrast, *DLK1* expression was not detected in HSC/MPP cells using the targeted or the WTA sequencing approach (Letter Fig. 3).

In the study by Zhang *et al.*, HSC marker genes such as *CRHBP*, *HOPX* and *MLLT3* were detected in the HSC clusters, with differential expression for *MLLT3* between HSC1 and HSC2 (Letter Fig. 4). However, no differential expression between these clusters were found for *HOPX* and *CRHBP* in the Zhang data, and *MPL*, the receptor for thrombopoietin, was almost undetected.

We report in our manuscript on the upregulation of *ADGRG6*, *HLA-E* and *DLK1* especially in HSC-1 cells. Also, in Zhang *et al.* these genes were overexpressed in HSC1 vs HSC2 clusters. However, *MMRNI* was expressed at low signals without any difference between HSC1 and 2, while in our study, we detect *MMRNI* expression being upregulated in HSC1. Two other interesting markers, *ATPIB1* and *GUCY1A1*, which we identified by our targeted approach, showed equally weak expression in HSC1 and 2 in the Zhang *et al.* data.

- 5) The comparison of our data using a targeted gene expression analysis with the WTA data from Zhang *et al.* demonstrates a higher dynamic range of gene expression quantification and increased sensitivity of low-expressed genes (eg. *MPL*) in our study. This becomes apparent for most HSC-associated genes, as exemplified in Letter Fig. 4. The higher dynamic range allows us to accurately detect subtle changes in gene expression upon differentiation induction and early lineage choice, using the tradeSeq analysis.

Letter Fig. 3. Expression of stem cell-related genes detected in HSPC populations from Triana *et al.* (three most immature clusters). A) Expression data of HSC marker genes from the targeted sequencing approach. B) Gene expression data from whole transcriptome analysis (WTA).

Komic *et al.*

Zhang *et al.*

Letter Fig. 4. Gene expression of HSC marker genes identified in our study (Komic *et al.*) and confirmed/compared by the study of Zhang *et al.*

Letter Fig. 4, continued. Gene expression of HSC marker genes identified in our study (Komic *et al.*) and confirmed/compared by the study of Zhang *et al.*

Study the immune-modulatory function of CD273/PD-L2 on HSPCs

Following the valuable recommendation by the Reviewer to “to delve deeper into the function of interesting findings such as CD273”, we functionally and molecularly investigated the immune-modulatory role of CD273/PD-L2 upregulated on immature HSPCs.

In the revised version of the manuscript, we have included a series of experiments with allogeneic T cell – HSPC cocultures from several donors (including PD-1/PD-L2 blocking), and these data are displayed in a new manuscript Figure 7. These experiments support an immune-regulatory function of HSPCs expressing PD-L2 in their capacity to decrease T cell proliferation, activation and pro-inflammatory cytokine release. After blocking PD-L2 on HSPCs, we also observed a reduced development of regulatory T cells (Treg).

These results add valuable functional information about the identified upregulation of surface PD-L2 expression on the most immature human HSPCs in modulating T cell activity.

Figure 7

Figure 7. Immune-modulatory function of CD273/PD-L2 on HSPCs

T-cells were co-cultured with allogeneic HSPCs in the presence of either PD-L2 blocking antibody (black) or an isotype control antibody (red) for 72 h, and compared to a T-cell mono-culture (blue). a) Proliferation of CD4⁺ and CD8⁺ T-cells by CFSE labeling (n=3). ANOVA with Tukey's multiple comparisons test. b) Proliferation index of CD4⁺ and CD8⁺ T-cells (n=3). ANOVA with Tukey's multiple comparisons test. c) Flow cytometry analysis of CD69 expression on CD8⁺ T-cells. d) Activation of CD8⁺ T-cells (n=4). ANOVA with Holm-Šídák's multiple comparisons test. e) T-cell subtype abundances relative to T-cell mono-culture (n=4). T_{regs} regulatory T-cells; T_N naïve T-cells, T_{CM} central memory T-cells; T_{EM} effector memory T-cells; T_{EMRA} terminally differentiated effector memory, T-cells re-expressing CD45RA. ANOVA with Holm-Šídák's multiple comparisons test. f) Cytokine-bead array assay of co-culture and T-cell mono-culture supernatants (n=3). ANOVA with Tukey's multiple comparisons test. Bars represent the mean with SEM. *, p<0.05; **, p<0.01; ***, p<0.001, ****, p<0.0001, n.s., not significant.

We added the following text to the Results section of the manuscript (lines 332ff):

CD273 functions as an immune-modulatory receptor in immature HSPCs

CD273 (PD-L2) binds to the receptor PD-1 on T-cells, known as an important immune checkpoint, to prevent excessive T-cell activation.³⁶ To investigate the potential immunomodulatory role of PD-L2 expression in the early hematopoietic stem/progenitor compartment, a mixed lymphocyte reaction (MLR) assay was established. CD34⁺ HSPCs and T-cells from four healthy human donors were MACS-enriched. Donor-individual HSPCs were co-cultured with unmatched T-cells isolated from another donor, leading to individual allogeneic pairs. T-cells were stimulated with beads coated with antibodies against CD2, CD3, and CD28, which trigger robust T-cell activation. The HSPCs were pre-treated either with a neutralizing CD273/PD-L2 antibody or with an IgG isotype control to evaluate the specific contribution of CD273/PD-L2 to immune modulation of T-cells. The co-cultures were then compared to control conditions, where T-cells were cultured separately.

To monitor T-cell proliferation dynamics in distinct T-cell subtypes over time during co-culture, in the presence and absence of CD273/PD-L2 neutralization, CFSE (CellTrace) and T-cell surface marker expression were determined by flow cytometry (Figure 7a) and quantified using the proliferation index (Figure 7b). The co-culture with HSPCs weakly suppressed the proliferation of CD8⁺ T-cells in comparison to T-cells alone. However, increased CD8⁺ T-cell proliferation was effectively rescued by blocking CD273/PD-L2. This observation suggests that HSPCs suppress CD8⁺ T-cell proliferation through PD-L2 interactions. In contrast, the proliferation of CD4⁺ T-cells was not altered by the MLR itself, indicating that CD4⁺ T-cells might not be as sensitive to HSPC-mediated suppression as CD8⁺ T-cells under these conditions. However, when CD273/PD-L2 was blocked, there was a marked increase in CD4⁺ T-cell proliferation. This suggests a delicate balance between T-cell stimulation and suppression within the co-culture environment, and that blocking CD273/PD-L2 may tip this balance towards stronger CD4⁺ T-cell activation.

To further determine the activation status of CD8⁺ T-cells, the expression of CD69, a marker of early T-cell activation³⁷, was assessed (Figure 7c). The co-culture showed a modestly non-significantly increased CD69 expression on CD8⁺ T-cells (Figure 7d). By contrast, CD273/PD-L2 inhibition resulted in an elevation of CD69 levels, reinforcing the idea that PD-L2 mediates a key suppressive signal for CD8⁺ T-cell activation.

Next, the effects of HSPC co-culture on memory T-cell populations were explored using flow cytometry (Figure 7e). The data showed a relative decrease in naïve T-cells (T_N) and central memory T-cell (T_{CM}) populations, while effector memory T-cells (T_{EM}) increased upon MLR, confirming that contact with HSPCs promotes T-cell specification towards more activated states. Of note, CD273/PD-L2 blockade resulted in a notable reduction of the fraction of CD4⁺CD25^{hi}FOXP3⁺ regulatory T-cells (T_{regs}). This result points to CD273/PD-L2 playing a dual role in immune suppression: (1) direct inhibition of T-cell activation through receptor-ligand interactions, and (2) indirect suppression via the induction and expansion of T_{regs}.

To substantiate phenotypic changes in T-cell subsets, T-cell function was evaluated by analyzing cytokine levels in the supernatants of the co-cultures using multi-cytokine assays (Figure 7f, Supplementary Figure S11). The T-cell activation cytokines IFN- γ , TNF- α and IL-6 were elevated in CD273/PD-L2 neutralized co-cultures compared to T-cell mono-cultures. The coculture of T-cells with HSPCs decreased the secretion of the pro-inflammatory cytokine IL-17A, while CD273/PD-L2 blocking markedly elevated IL-17A levels. These findings strengthen the hypothesis that the interaction of CD273/PD-L2 promotes an immunosuppressive environment, leading to reduced T-cell activation and inflammation.

We added the following section to the Discussion (lines 452ff):

HSCs have been proposed to have immune modulatory functions due to expression of MHC class II molecules that may allow HSCs to present antigens influencing T-cell responses⁵³. Lately, Beta-2-microglobulin expression on HSCs has been proposed as a crucial "don't eat-me" signal in zebrafish, regulating the HSC-macrophage interaction and HSC clonality.⁵⁴ However, the expression of PD-1 ligands may serve other purposes than protecting HSCs against auto-reactive T-cells. In our study, we could confirm a higher surface expression of CD273/PD-L2 on a subset of human HSCs by flow cytometry. Prospectively isolated HSCs with high CD273 expression were found to display delayed differentiation, multi-lineage potential and enhanced quiescence, all attributes important for HSC physiology. Increased expression of stemness genes suggests that potent human HSCs may reside in the CD273^{high} HSC fraction, which needs adequate functional *in vivo* assessment. Here we demonstrated an immune-modulatory function of CD273/PD-L2 expressed by HSPCs. Allogeneic T-cell proliferation and activation were elevated upon CD273/PD-L2 blockade. Furthermore, the release of pro-inflammatory cytokines increased. Lastly, T-cell differentiation into regulatory T-cells (T_{reg}) was reduced after CD273/PD-L2 inhibition on HSPCs. These results suggest that CD273/PD-L2 plays an immunosuppressive role on HSCs, preventing excessive T-cell activation and potential auto-reactivity. The data highlight

CD273/PD-L2 as a safeguard for immature HSC populations, acting as both a direct inhibitor of T-cell responses and as a facilitator of T_{reg}-mediated immune regulation. These findings provide valuable insights into the complex immune interactions between HSCs and T-cells, and underscore the importance of CD273/PD-L2 in maintaining immune homeostasis.

Summary of findings:

The recommendation of both Reviewers to compare our data with the previously published studies of Triana *et al.* and Zhang *et al.* was very helpful. These comparisons highlight the following positive aspects of our study:

- Higher number of immature HSPCs to resolve subclusters of cells (see Letter Table 1)
- Inclusion of more individuals (n=15) representing three age groups (see Letter Table 1)
- Rational selection of genes relevant for stemness and early differentiation induction
- Targeted sequencing approach for a higher dynamic range of gene expression quantification of even low expressed genes
- Reanalysis of the published data sets confirmed some of our reported genes found to be overexpressed in HSC1
- Our study revealed novel marker genes not reported or detected in the published data sets.
- Identification of CD273/PD-L2 as an immune-modulatory receptor of T-cell activation high expressed on most immature human HSPCs

We added Supplementary Table 8 comparing the marker genes of the most immature HSPC cluster in each of the studies Komic *et al.*, Zhang *et al.* and Triana *et al.*

We added the following sentence to the Results section (lines 182ff):

We confirmed increased expression of *DLK1* and *ADGRG6* in the most immature HSPC clusters of previously published single cell expression datasets (Supplementary Table 8).^{8,13}

8. Triana, S. *et al.* Single-cell proteo-genomic reference maps of the hematopoietic system enable the purification and massive profiling of precisely defined cell states. *Nature immunology* **22**, 1577–1589; 10.1038/s41590-021-01059-0 (2021).
13. Zhang, X. *et al.* An immunophenotype-coupled transcriptomic atlas of human hematopoietic progenitors. *Nature immunology* **25**, 703–715; 10.1038/s41590-024-01782-4 (2024).

Line 474, the author claim they found several new candidate markers for some cell states, please make sure to check their expression in published bone marrow atlas. I'm pretty sure HLF has been reported numerous times.

#Reply 2: Thank you for raising this comment. We checked all mentioned markers on their novelty in the context of hematopoietic lineage specification and corrected our list accordingly. HLF has been reported to be up-regulated in HSCs before, and we corrected this point in our manuscript.

The authors need to change CITE-seq to Abseq to avoid confusion. Cellular Indexing of Transcriptomes and Epitopes by Sequencing (CITE-seq) is a multimodal single cell phenotyping method developed in the Technology Innovation lab at the New York Genome Center. CITE-seq uses 10X 3' or 5' kit (whole transcriptome, 10X genomics) and ADTs (Biolegend) where Abseq the authors used is a targeted transcriptome plus antibody method developed by BD Rhapsody (well-based). Check Simon Hass lab 2021 Nature Immunology paper for reference. Also, it should be tradeSeq instead of TRADE-seq, make sure to get all these nomenclatures right.

#Reply 3: Thank you for raising these points. TRADE-seq has been changed to tradeSeq throughout the manuscript. Regarding CITE-seq, we have noted a shift in nomenclature. Accordingly, BD is using the terminology of CITE-seq in their information material of the BD Rhapsody System describing the Cellular Indexing of Transcriptomes and Epitopes by Sequencing, thus it seems likely that the term may have been standardized when using combined transcriptome and surface protein analysis at a single cell level, regardless of the platform used. We would prefer to use the term CITE-Seq since most literature using a proteo-genomic single cell technology is searchable by the expression "CITE-Seq". However, if the reviewer or the editors prefer changing the wording to "Abseq" we are happy to do so.

Secondly, pseudotime, trajectory analysis and to larger extend, scRNA-seq provides hypothesis and should be carefully examined and validated by functional readout. A stem cell is define based on its self-renewal and multipotency not by just a few gene expression. Hypothesis such as difference of cell differentiation or fate in this study need to be validated by flow cytometry (accumulation of certain immunophenotypic defined progenitor) and colony formation assays to examine lineage output (CFU). Specially, CD273⁺ as a stem cell marker, can CD34⁺CD273⁺ cells self-renew and give rise all lineages like CD34⁺CD38⁻CD90⁺CD45RA⁻ HSC?

The author cultured sorted CD273^{hi} cells for 7 days in SFEM II with cytokines and then were flow profiled for surface markers, checked for gene expression and cycle entry delay. These are good clues but insufficient to profile a stem cell without functional readout such as long-term culture, CFU to exam lineage output, and xenograft.

#Reply 4: We agree with the Reviewer that the link of functional data with single cell molecular data is paramount to determine the function and potency of cells. This was the reason why we performed the functional tests of FACS-sorted CD273^{hi} and CD273^{low} CD34⁺CD38⁻CD45RA⁻Lin⁻

shown in the original version of the manuscript (Figure 6: *in vitro* differentiation culture, time-lapse -based cell tracking, *in vitro* expansion).

Guided by the Reviewer's comment, we performed additional experiments of a serial replating colony-formation unit assay to read out colony formation and multi lineage potential *in vitro*. We compared CD273^{hi} and CD273^{low} CD34⁺CD38⁻CD45RA⁻Lin⁻ with CD90⁺CD49f⁺CD34⁺CD38⁻CD45RA⁻Lin⁻ HSPCs as recommended by the Reviewer. The data is presented in new Figure 6h and 6i and Supplementary Figure 10 and the graphs are shown below.

Refer to Fig. 6h and i.

h

i

Refer to Suppl. Fig. 10a and b

We show that CD273^{hi}CD34⁺CD38⁻CD45RA⁻Lin⁻ have GEMM potential *in vitro*. Replating efficiency was similar between the tested HSPC populations. Here we demonstrate multi-lineage potential of CD273^{hi} CD34⁺CD38⁻CD45RA⁻Lin⁻ HSPCs and *in vitro* replating capacity, which is not different to the other tested HSPC populations. Unfortunately, given the limited revision time, it was not possible to measure self-renewal as long-term reconstitution potential *in vivo* after serial transplantation.

We added the following sentences to the Results section of the manuscript (lines 325):

Last, we performed colony-forming unit (CFU) assays with serial replating to demonstrate multi-lineage differentiation potential. We compared CD273^{high} and CD273^{low} HSPCs with CD90⁺CD49f⁺ HSCs. We did not observe any difference in CFU ability and numbers, myeloid lineage differentiation or serial replating capacity between these populations, demonstrating that CD273^{high} HSPCs have equal – but not superior *in vitro* capacity as CD90⁺CD49f⁺ HSCs (Figure 3h-i, Supplementary Figure S10) ⁴.

We added the following sentences to the Methods section of the manuscript (Lines 693ff):

Colony-forming unit assay

Primary colony-forming unit assays (CFUs) were performed by seeding 300 FACS-sorted cells from the following populations: lin-CD34⁺CD38⁻CD45RA⁻, lin-CD34⁺CD38⁻CD45RA⁻CD273^{high}, lin-CD34⁺CD38⁻CD45RA⁻CD273^{low} and lin-CD34⁺CD38⁻CD45RA⁻CD90⁺CD49f⁺ in 3 ml of MethoCult™ medium (H4034, STEMCELL Technologies). From this, 1 ml—equivalent to approximately 100 cells per plate—was plated into each 35 mm dish. Plates were incubated at 37°C with 5% CO₂ for 14 days, after which colony formation and lineage distribution were assessed microscopically (CellObserver, Zeiss). For the secondary assay, cells were harvested from CFUs and counted. For each sample, 30,000 cells were resuspended in 3 ml MethoCult™ and divided into plates at 10,000 cells per plate. Secondary colonies were then scored after an additional 14 days of incubation at 37°C with 5% CO₂.

Finally, data transparency. Data and code are not available, and many statistical/biological details and supplementary data are missing, making it impossible to judge the rigor of this study.

Data availability is not in line with Nature journals joint policies. The authors should provide reviewer link to the sequencing data repository instead the asking the editor to request access. Also, all code should be readily available on public platforms eg. github at the time of review instead of “available upon a scientifically reasonable request”. Flow data is recommended to be uploaded to flowrepository.

#Reply 5: At the time of original submission to the journal, we stored our primary data at Zenodo (with access upon request from the editors) and we published our codes on Github with open access (<https://github.com/TessaSchm/Early-HSC-differentiation/tree/main>). However, this information was accidentally not sent to the Reviewers in time. We deeply apologize for the confusion and the difficulty to access and review our data sets.

For the revised version of our manuscript, we followed the recommendation of Reviewer #2 not to use Zenodo but we share our primary data on ArrayExpress. Raw count matrices, sample tag calls, rds files and detailed sample information files can be accessed using the following link:

<https://www.ebi.ac.uk/biostudies/arrayexpress/studies/E-MTAB-14596?key=6eb62c54-2e2a-466e-bc78-a2c690645830>

Our R codes are openly available under:

<https://github.com/TessaSchm/Early-HSC-differentiation/tree/main>

Current supplementary information is not adequate nor transparent that the reviewer or potential readers could examine the data or reproduce their findings. For instances, author should provide a supplementary table to list all samples by age, sex and race, and also the cell frequencies in each. A supplementary table to list all cell barcode to cell annotations and correlation score. Supplementary tables to list differentially regulated surface marker expression. Authors should also explain what cutoff is used to call doublets, cell filtering and assigning cells to each sample tag, and provide processed filtered matrix as raw data (not possible to see if they are deposited without access) or excel sheet. Furthermore, the statistical significance or parameters are missing, for example How was label transfer performed, based on what method and cutoff?

#Reply 6: Thank you very much for listing the required meta data (sample origin, cell identifiers, label transfer, and filter conditions), which have been included now in our revised version of the manuscript. Supplementary Tables 3, 5, 8, 13, 15 with sample information, cell barcodes, and differentially regulated surface markers expression have been added. More information on data filtering is provided below and a summary of this information has been included in the Methods section of the revised manuscript, lines 534ff.

New Supplementary Tables:

- Suppl. Table 3: Donor information (CITE-seq)
- Suppl. Table 5: Cluster annotation for CD34⁺ UMAP (manual and cell label transfer from Triana et al.)
- Suppl. Table 8: Comparison of upregulated marker genes in the most immature cell cluster of three independent datasets (genes identified by FindAllMarkers)
- Suppl. Table 13: Differential surface protein expression of HSC-1 vs. HSC-2
- Suppl. Table 15: Donor information for functional and molecular validation experiments

Cell calling: The number of mRNA reads of each cell is plotted on a log10-transformed cumulative curve, with cells sorted by the number of reads in descending order. Additionally, the rate of change of the cumulative count is calculated with the second derivative. A distinct inflection point observed in the second derivative, indicating a division between signal cell labels and noise cell labels is taken as a threshold. The second derivative is determined for each dataset and thresholds are applied according to second derivative inflection point.

Assigning Cells to Sample Tag: A high-quality singlet was defined as putative cell where more than 75% of Sample Tag reads are from a single tag. When a singlet is identified, the counts for all the other tags were considered Sample Tag noise. The minimum Sample Tag read count for a putative cell to be positively identified with a Sample Tag is defined as the lowest read count of a high-quality singlet for this Sample Tag. To improve sample determination and recover singlets that are not initially considered high quality, the algorithm subtracts the expected number of per-cell noise counts from each Sample Tag. The total expected per-cell noise, derived from the trend line, is multiplied by the percentage of noise contribution of each Sample Tag to determine the expected noise per Sample Tag. After subtracting the expected per tag noise, any Sample Tag that has a count higher than its minimum read count is called for that cell, and the putative cell is considered a called cell.

Determining the multipliers: When the counts of two or more Sample Tags exceed their minimum thresholds, then that putative cell is called as a cross-sample Multiplier, indicating more than one actual cell in the microwell, and the cells are of different samples of origin (ref doc. for above: <https://www.bdbiosciences.com/content/dam/bdb/marketing-documents/products-pdf-folder/software-informatics/rhapsody-sequence-analysis-pipeline/Rhapsody-Sequence-Analysis-Pipeline-UG.pdf>)

Cell label transfer: All clusters were initially manually annotated using previously published papers and dataset, mostly relying on well-known marker genes and cell surface proteins. To confirm our annotations, we utilized existing WTA dataset of full bone marrow cells published by Triana et al. The reference dataset was log normalized for mRNA counts, scaled using the ScaleData function, followed by reduction PCA by employing RunPCA function.

FindTransferAnchors function was used to find anchors for each Seurat object ($k = 30$), where “Prediction_HCA” labels were used to transfer names to the query dataset using TransferData function. The predictions were added to metadata under the name “predictions_mrna.” There was no cut off value selected.

The authors should explain by what statistical standard did they use to select “596 highly relevant genes for human hematopoiesis”. To defend such selection, the authors should rigorously examine their data complexity and cell annotations with full single-cell transcriptome datasets to make sure the targeted transcriptome is equivalent and cell composition is comparable. For instance, there is a more comprehensive (>300k single cells) CITE-seq atlas of human bone marrow CD34+ progenitors in a Nature Immunology paper by Lee Grimes lab earlier this year.

Reply 7: Thank you for asking how we rationally selected for genes in our panel, which is key for a targeted gene expression approach. The main aim of our study was to resolve early gene expression changes between immature HSCs and early multipotent progenitor populations, and to

resolve heterogeneity within the immature HS(P)Cs. The core part of the genes was chosen based on the hallmark study by Ebert et al. based on bulk RNA sequencing of FACS-purified HSC, LSC and Progenitor cell populations (Eppert et al. *Nat. Med* 2011). Second, we further enriched for genes in the LSC17 signature of leukemia stem cells (Ng et al. *Nature* 2016, van Galen et al. *Cell* 2019) and in genes often mutated in clonal hematopoiesis (a clonal advantage established at HSC level). We included genes encoding for hematopoietic surface markers and immune modulatory receptors (Giustacchini et al. *Nat Med* 2017). Furthermore, cell cycle reporter genes were added. In conclusion, we did not intend to accurately reconstruct the whole hematopoietic development of all mature lineages by a targeted sequencing approach, since we aimed at investigating the early differentiation induction of HSCs and the regulation of gene expression at immature HSPCs along their earliest lineage branching points.

The antibodies used for CITE-Seq covered surface markers of immature HSPC subpopulations, oligo- and unipotent progenitors and mature blood cell types of all lineages, further delineating cell identification and lineage specification.

We extended the explanation of our rationale to select the genes in our panel in the manuscript Results section, lines 93ff.

To resolve cellular heterogeneity and gene expression dynamics in early differentiation steps of human BM HSPCs, we rationally selected 596 genes for deep-targeted CITE-seq using the Rhapsody technology (Supplementary Table 1). First, genes previously reported to be differentially expressed in bulk-purified human HSCs and progenitors were selected¹⁸. Second, we included genes expressed in leukemia stem cells¹⁸⁻²⁰ and mutated in clonal hematopoiesis and myeloid neoplasms²¹. We further added hematopoietic surface marker genes, immune modulatory receptors, and cell cycle reporter genes to our panel^{8,22}. The 46 antibodies used to capture the surface proteome of the same cells covered surface markers of immature HSPC subpopulations, oligo- and unipotent progenitors, mature blood cell types and immune receptors, further delineating cell identification and lineage specification (Supplementary Table 2)^{4,23,24}.

Furthermore, we added heatmaps showing the correlation between our cell clusters (UMAP from Figure 1) and the cell labels from the Triana *et al.* dataset and the WTA dataset from Zhang *et al.*, confirming high concordance of cell identities between these studies (Supplementary Figure S5b, Letter Figure 5).

A Correlation with Triana *et al.*

B Correlation with Zhang *et al.*

Letter Fig. 5. Correlation matrices with previously published BM datasets.

These matrices show a high concordance in the cluster annotation of our study (y-axis) and of the study of Triana *et al.* (x-axis, A) or the study of Zhang *et al.* (B).

Line 505, mobilized CD34+ cells and bone marrow cells could be very different and should not be used to confirm findings from bone marrow (niche).

#Reply 8: Thank you for raising this important point. We agree that HSPCs from bone marrow and from peripheral blood may represent different cell identities and show distinct properties and behavior. Therefore, we carefully compared the molecular features and expression profiles of HSPCs sorted from bone marrow and peripheral blood from the same donors isolated at the same day by our proteo-genomic BD Rhapsody approach. We analyzed paired BM and PB samples from five healthy donors used in our dataset. We generated pseudobulk per individual and per compartment and the plot below shows a principal component analysis (PCA) (Letter Figure 6). We found high correlation of the single cell data obtained from BM and PB of the same donor, and more variance between individual donors, suggesting that the complex single cell expression data generated from CD34+ cells is highly comparable.

Letter Fig. 6. Principal component analysis (PCA) of CITE-Seq data of CD34+-enriched HSPCs from BM and peripheral blood (PB) of the same donor. 5 donors providing BM and PB samples at the same day were included.

Second, we looked at the distribution of CD273+ cells in the HSPC population (CD34+CD38-CD45RA-lin-) from BM donors and mobilized blood donors by FACS (n= 7 donors per source). Again, we found similar percentages of CD273-expressing cells in HSPCs between BM and mobilized peripheral blood (unrelated donors), Supplementary Figure S9b.

Both results suggest that the populations we functionally investigated in our study show similar features in BM and PB.

Suppl. Fig. 9b. CD273 expression on CD34+CD38-CD45RA- HSPCs from bone marrow (BM) and mobilized peripheral blood (mPB), determined by flow cytometry. n= 7 donors per source. Mean with SEM is displayed. No significant difference.

Line 509, it is unclear what flow cytometry gate was used and how was it determined for sorting CD273 high and CD273 low cells (assuming both express CD273?). The number of cells in each gate is critical but missing. Judging by the flow data in Figure S9, CD273 positive cells seems very few (0.1%?) which makes me wonder if the cell frequency you see in flow match what's in the Abseq. This information is very important to evaluate whether CD273 can be used as good flow marker.

Reply 9: We agree with the reviewer that this was not entirely clear in the original submission. We have thus prepared a new supplementary figure replacing Supplementary Figure S9. Here we show the staining and gating for CD273⁺ CD34⁺CD38⁻CD45RA⁻Lin⁻ in bone marrow and in mobilized blood from n= 7 donors for each source. There was no difference in the frequency of CD273⁺ cells between both sources, which was at about 5% of CD34⁺CD38⁻CD45RA⁻Lin⁻ HSPCs.

We added the following sentence to the Results section of the manuscript (lines 299ff):

About 5% of CD34⁺CD38⁻CD45RA⁻Lin⁻ HSPCs express high levels of CD273 in bone marrow and peripheral blood (Supplementary Figure S9).

a**b**
Supplementary Figure S9. Gating strategy for FACS sorting of CD273^{high} and CD273^{low} HSCs. a) Dot-plots illustrating the hierarchical gating strategy used for isolation of CD34+CD38-CD45RA- HSPCs according to their CD273 expression. b) Bar diagram showing no difference between mobilized PB and BM CD273 content.

Minor concerns,

Line 275, citation errors

#Reply 10: Corrected. Thank you for bringing it up.

Line 323, “Forty-six oligo nucleotide-labeled antibodies commonly used to characterize HSPCs and hematopoietic cells by flow cytometry were included to capture the surface proteome of the same cells.” Add rationale or citation for “commonly”.

#Reply 11: We included respective citations to explain our selection of antibodies against surface markers used for flow cytometry-based analysis/sorting of stem/progenitor populations and mature blood cell lineages, as well as NK / T-cell ligands, which are well-established and widely applied in the field.

We included the following sentence to the Results section (lines 99ff):

The 46 antibodies used to capture the surface proteome of the same cells covered surface markers of immature HSPC subpopulations, oligo- und unipotent progenitors, mature blood cell types and immune receptors, further delineating cell identification and lineage specification (Supplementary Table 2)^{4,23,24}.

4. Notta, F. *et al.* Isolation of single human hematopoietic stem cells capable of long-term multilineage engraftment. *Science (New York, N.Y.)* **333**, 218–221; 10.1126/science.1201219 (2011).
23. Notta, F. *et al.* Distinct routes of lineage development reshape the human blood hierarchy across ontogeny. *Science (New York, N.Y.)* **351**, aab2116; 10.1126/science.aab2116 (2016).
24. Sivori, S. *et al.* Human NK cells: surface receptors, inhibitory checkpoints, and translational applications. *Cellular & molecular immunology*, 1–12; 10.1038/s41423-019-0206-4 (2019).

Line 365, remove “comprehensive” here and its appearance in most cases. 60,000 cells of human CD34+ is very small compared to studies of recent years and most immature stem cells recognized as human “HSC” (CD34+CD38-CD90+CD45RA-) is only 2% of CD34+ cells (~1200).

#Reply 12: We agree that we should not use priority terms in our manuscript, so we deleted “comprehensive”.

Figure 6, need to specify cells from which donor is used for each panel, are they donor matched?

#Reply 13: In total, we used cells from 33 different donors for our functional and molecular studies shown in Figures 6 and 7. The information about the donors (sex, age) is provided as Supplementary Table 15, including the performed experiments. We did not use matched donors for different substudies since most experiments required most of the cells, and primary cell material was limited.

Reviewer #1 (Remarks on code availability):

code is not readily available to the reviewer.

Please see our Reply #5

Reviewer #2 (Remarks to the Author):

This manuscript systematically investigates the early gene network changes in human hematopoietic stem cells (HSCs) during differentiation using a targeted single-cell CITE-seq approach on FACS-enriched bone marrow stem and progenitor cells from 15 healthy donors. Their analyses show young donors exhibited more productive differentiation from HSCs to committed progenitors. Conversely, the study reveals a delayed differentiation to all lineages in the elderly, with a pronounced effect in the lymphoid lineage. This finding could help explain the overabundance of HSCs in the elderly, accompanied by a myeloid shift in further differentiated cells. TRADE-Seq analysis showed continuous gene expression changes, highlighting both known HSC-related genes and new immature marker genes (DLK1, ADGRG6), providing a gene expression roadmap at the earliest branching points. The study also identified CD273 (PD-L2) as highly expressed in the most immature stem cells. The study demonstrates strength in the accurate quantification of lowly expressed genes. This is achieved through the targeted approach of deep scCITE-seq, which comes with a low dropout rate for the targeted genes. This paper is overall well written, although more structuring of the discussion section (which has a very long initial paragraph) would be helpful.

#Reply 1: Thank you very much for your positive comments and constructive feedback. We have restructured the discussion section and condensed the first paragraph to its essential points. Furthermore, as described above in response to Reviewer 1, we have included additional experiments with allogenic T cell – HSPC cocultures and have thus extended the discussion on the immune-modulatory functions of HSPCs. We hope that the restructured discussion is better focused and clearly based on the main findings of our study.

The manuscript lacks clarity regarding the motivation for assessing the expression of genes associated with clonal haematopoiesis (CH). Furthermore, the interpretation of the outcomes related to these findings is not adequately addressed, resulting in a disjointed paragraph that appears without sufficient context. Were any CH mutations found in the scCITE-seq data? Or would these be 'invisible' to the targeted approach taken?

#Reply 2: Thank you for alerting us about the lack of explanation to show CH-related genes and LSC-related genes in our manuscript. This point should indeed be better explained and the rationale behind this analysis should become obvious in our revised manuscript.

The gene panel was rationally designed to cover expressed genes in hematopoietic populations within the CD34⁺ HSPC progenitors, and particularly focused on enriching genes being expressed in immature HSPCs (HSCs) and at early stages of HSC differentiation. Since single somatic mutations in leukemia-related genes cause clonal dominance and expansion in HSCs leading to detectable clonal hematopoiesis over time, we were particularly interested in the expression pattern

of 56 CH-related genes and of genes of the acute leukemia stem cell signature. This is of particular importance when therapies are directly targeting these leukemia-drivers (e.g. *JAK2*, *IDH1/2*), and a dependence of these gene products at stem cell and leukemia stem cell level has been debated. Therefore, we included these genes in our targeted panel (Dorsheimer et al. *JAMA Cardiol* 2019; Ng et al. *Nature* 2016) in order to quantitate gene expression of these genes in immature HSPCs and their progeny at earliest differentiation induction. Mechanistic consequences could be explained by the expression of somatically mutated genes in either HSCs or early MPPs (e.g. increased self-renewal, delay in differentiation).

Since the BD Rhapsody technology is based on short read 3'-sequencing, mutation calling for most CH-related mutations is not possible. Therefore, we do not have the information about the CH mutation status, which would primarily affect the aged donors. Unfortunately, we did not request permission from the donors or the ethics board to acquire the CH mutation status of the donors included in this study.

We added the following introductory chapter to the Results section, lines 202ff:

Individual somatic mutations in genes associated with myeloid neoplasms in HSCs can cause the emergence of clonal hematopoiesis (CH)²¹. The mechanism of clonal dominance in HSCs is dependent on the affected gene and its expression pattern during hematopoietic differentiation^{32,33}. Furthermore, the expression of (mutated) leukemia-associated driver genes in leukemia-initiating stem cells and healthy HSCs has consequences for targeted therapies. Therefore, we included genes associated with CH and myeloid neoplasms in our targeted gene panel and investigated their expression dynamics in cells from clusters HSC-1, HSC-2 and MPP (Figure 2g).

While *DLK1* and *ADGRG6* could be relevant immature marker genes further functional validations are necessary to better understand the roles of these markers in HSC biology.

#Reply 3: We agree with the Reviewer that our findings open important new avenues for functional validation studies of the role of these HSC1-upregulated marker genes such as *DLK1* and *ADGRG6*. We also agree that both genes and their gene products are very interesting candidates to influence human HSC behavior and physiology.

In our manuscript, we discuss their potential function by referring to the current literature. For *DLK1*, a mouse study shows that the absence of *DLK1* leads to reduced HSC repopulation upon transplantation³⁰.

30 Huang, D. *et al.* *Dlk1* maintains adult mice long-term HSCs by activating Notch signaling to restrict mitochondrial metabolism. *Experimental Hematology & Oncology* **12**; 10.1186/s40164-022-00369-9 (2023).

ADGRG6 encodes for a G-protein-coupled surface receptor GPR126. Its function in hematopoiesis has not been studied. However, expression studies showed an upregulation of *ADGRG6* in a subset of acute myeloid leukemias, which is linked to worse prognosis⁴³. This was the reason why we added this gene to the targeted sequencing approach.

43. Maiga, A. *et al.* Transcriptome analysis of G protein-coupled receptors in distinct genetic subgroups of acute myeloid leukemia: identification of potential disease-specific targets. *Blood cancer journal* 6, e431; 10.1038/bcj.2016.36 (2016).

Based on our findings reported in the manuscript, we will focus our future studies on untangling the role of these two molecules in human hematopoiesis and leukemia, including CRISPR/Cas 9 knock-out and lentiviral overexpression in primary human HS(P)Cs, followed by rigorous functional analysis in *in vitro* and *in vivo* transplantations. Unfortunately, the time needed to perform such experiments precludes the possibility to add it in the revised version of the manuscript. Instead, we functionally investigated the immune-modulatory role of another protein identified to be high expressed in the most immature HSPC cluster in our study, CD273/PD-L2, also motivated by the recommendation of Reviewer 1.

In the revised version of the manuscript, we have included a series of experiments with allogeneic T cell – HSPC cocultures from several donors (including PD-1/PD-L2 blocking), and these data are displayed in a new manuscript Figure 7. These experiments support an immune-regulatory function of HSPCs expressing PD-L2 in their capacity to decrease T cell proliferation, activation and pro-inflammatory cytokine release. After blocking PD-L2 on HSPCs, we also observed a reduced development of regulatory T cells (Treg).

These results add valuable functional information about the identified upregulation of surface PD-L2 expression on the most immature human HSPCs in modulating T cell activity.

Figure 7

Figure 7. Immune-modulatory function of CD273/PD-L2 on HSPCs

T-cells were co-cultured with allogeneic HSPCs in the presence of either PD-L2 blocking antibody (black) or an isotype control antibody (red) for 72 h, and compared to a T-cell mono-culture (blue). a) Proliferation of CD4⁺ and CD8⁺ T-cells by CFSE labeling (n=3). ANOVA with Tukey's multiple comparisons test. b) Proliferation index of CD4⁺ and CD8⁺ T-cells (n=3). ANOVA with Tukey's multiple comparisons test. c) Flow cytometry analysis of CD69 expression on CD8⁺ T-cells. d) Activation of CD8⁺ T-cells (n=4). ANOVA with Holm-Šídák's multiple comparisons test. e) T-cell subtype abundances relative to T-cell mono-culture (n=4). T_{regs} regulatory T-cells; T_N naïve T-cells, T_{CM} central memory T-cells; T_{EM} effector memory T-cells; T_{EMRA} terminally differentiated effector memory, T-cells re-expressing CD45RA. ANOVA with Holm-Šídák's multiple comparisons test. f) Cytokine-bead array assay of co-culture and T-cell mono-culture supernatants (n=3). ANOVA with Tukey's multiple comparisons test. Bars represent the mean with SEM. *, p<0.05; **, p<0.01; ***, p<0.001, ****, p<0.0001, n.s., not significant.

We added the following text to the Results section of the manuscript (lines 332ff):

CD273 functions as an immune-modulatory receptor in immature HSPCs

CD273 (PD-L2) binds to the receptor PD-1 on T-cells, known as an important immune checkpoint, to prevent excessive T-cell activation.³⁶ To investigate the potential immunomodulatory role of PD-L2 expression in the early hematopoietic stem/progenitor compartment, a mixed lymphocyte reaction (MLR) assay was established. CD34⁺ HSPCs and T-cells from four healthy human donors were MACS-enriched. Donor-individual HSPCs were co-cultured with unmatched T-cells isolated from another donor, leading to individual allogeneic pairs. T-cells were stimulated with beads coated with antibodies against CD2, CD3, and CD28, which trigger robust T-cell activation. The HSPCs were pre-treated either with a neutralizing CD273/PD-L2 antibody or with an IgG isotype control to evaluate the specific contribution of CD273/PD-L2 to immune modulation of T-cells. The co-cultures were then compared to control conditions, where T-cells were cultured separately.

To monitor T-cell proliferation dynamics in distinct T-cell subtypes over time during co-culture, in the presence and absence of CD273/PD-L2 neutralization, CFSE (CellTrace) and T-cell surface marker expression were determined by flow cytometry (Figure 7a) and quantified using the proliferation index (Figure 7b). The co-culture with HSPCs weakly suppressed the proliferation of CD8⁺ T-cells in comparison to T-cells alone. However, increased CD8⁺ T-cell proliferation was effectively rescued by blocking CD273/PD-L2. This observation suggests that HSPCs suppress CD8⁺ T-cell proliferation through PD-L2 interactions. In contrast, the proliferation of CD4⁺ T-cells was not altered by the MLR itself, indicating that CD4⁺ T-cells might not be as sensitive to HSPC-mediated suppression as CD8⁺ T-cells under these conditions. However, when CD273/PD-L2 was blocked, there was a marked increase in CD4⁺ T-cell proliferation. This suggests a delicate balance between T-cell stimulation and suppression within the co-culture environment, and that blocking CD273/PD-L2 may tip this balance towards stronger CD4⁺ T-cell activation.

To further determine the activation status of CD8⁺ T-cells, the expression of CD69, a marker of early T-cell activation³⁷, was assessed (Figure 7c). The co-culture showed a modestly non-significantly increased CD69 expression on CD8⁺ T-cells (Figure 7d). By contrast, CD273/PD-L2 inhibition resulted in an elevation of CD69 levels, reinforcing the idea that PD-L2 mediates a key suppressive signal for CD8⁺ T-cell activation.

Next, the effects of HSPC co-culture on memory T-cell populations were explored using flow cytometry (Figure 7e). The data showed a relative decrease in naïve T-cells (T_N) and central memory T-cell (T_{CM}) populations, while effector memory T-cells (T_{EM}) increased upon MLR, confirming that contact with HSPCs promotes T-cell specification towards more activated states. Of note, CD273/PD-L2 blockade resulted in a notable reduction of the fraction of CD4⁺CD25^{hi}FOXP3⁺ regulatory T-cells (T_{regs}). This result points to CD273/PD-L2 playing a dual role in immune suppression: (1) direct inhibition of T-cell activation through receptor-ligand interactions, and (2) indirect suppression via the induction and expansion of T_{regs}.

To substantiate phenotypic changes in T-cell subsets, T-cell function was evaluated by analyzing cytokine levels in the supernatants of the co-cultures using multi-cytokine assays (Figure 7f, Supplementary Figure S11). The T-cell activation cytokines IFN- γ , TNF- α and IL-6 were elevated in CD273/PD-L2 neutralized co-cultures compared to T-cell mono-cultures. The coculture of T-cells with HSPCs decreased the secretion of the pro-inflammatory cytokine IL-17A, while CD273/PD-L2 blocking markedly elevated IL-17A levels. These findings strengthen the hypothesis that the interaction of CD273/PD-L2 promotes an immunosuppressive environment, leading to reduced T-cell activation and inflammation.

We added the following section to the Discussion (lines 452ff):

HSCs have been proposed to have immune modulatory functions due to expression of MHC class II molecules that may allow HSCs to present antigens influencing T-cell responses⁵³. Lately, Beta-2-microglobulin expression on HSCs has been proposed as a crucial "don't eat-me" signal in zebrafish, regulating the HSC-macrophage interaction and HSC clonality.⁵⁴ However, the expression of PD-1 ligands may serve other purposes than protecting HSCs against auto-reactive T-cells. In our study, we could confirm a higher surface expression of CD273/PD-L2 on a subset of human HSCs by flow cytometry. Prospectively isolated HSCs with high CD273 expression were found to display delayed differentiation, multi-lineage potential and enhanced quiescence, all attributes important for HSC physiology. Increased expression of stemness genes suggests that potent human HSCs may reside in the CD273^{high} HSC fraction, which needs adequate functional *in vivo* assessment. Here we demonstrated an immune-modulatory function of CD273/PD-L2 expressed by HSPCs. Allogeneic T-cell proliferation and activation were elevated upon CD273/PD-L2 blockade. Furthermore, the release of pro-inflammatory cytokines increased. Lastly, T-cell differentiation into regulatory T-cells (T_{reg}) was reduced after CD273/PD-L2 inhibition on HSPCs. These results suggest that CD273/PD-L2 plays an immunosuppressive role on HSCs, preventing excessive T-cell activation and potential auto-reactivity. The data highlight

CD273/PD-L2 as a safeguard for immature HSC populations, acting as both a direct inhibitor of T-cell responses and as a facilitator of T_{reg}-mediated immune regulation. These findings provide valuable insights into the complex immune interactions between HSCs and T-cells, and underscore the importance of CD273/PD-L2 in maintaining immune homeostasis.

Line 441: The text states that there was no reduction of LYP, which does not appear to be in line with the remainder of their results presented. Clarification is therefore needed: did the authors find a reduction or not?

#Reply 3: Thank you for asking us to clarify this point. We did not see a significant reduction of the LYP upon aging. We agree that this statement in the manuscript was not clear, so we have corrected it in the Results section (Lines 228ff).

“Interestingly, we did not find a reduction of cells in LYP upon aging, which would explain the myeloid bias of hematopoiesis seen in older individuals. On the contrary, we even observed a reduced percentage of GMP and MDP1 and 2 in BM from aged individuals, in comparison to young donors.”

Line 400: The identification of new immature marker genes such as *ADGRG6* raises the question whether the authors can find corroborating evidence from other datasets, including published expression data?

Reply 4: Thank you for this important suggestion to consolidate our findings of HSC1-uperegulated marker genes by published datasets. We re-analyzed the primary sequencing data provided by two well-known studies in the field, Triana *et al.* Nat. Immunol 2021 and Zhang *et al.* Nat. Immunol 2024, to confirm HSC marker gene expression. Indeed, both data sets show increased expression of *ADGRG6* in HSC1 vs HSC2 (Zhang *et al.*, whole transcriptome analyses) and in HSC/MPPs vs. lymphomyeloid (LMP) and erythro-myeloid (EMP) progenitors (Triana *et al.*, whole transcriptome analysis) (Letter Figure 7).

Triana, S. *et al.* *Nature immunology* **22**, 1577–1589; 10.1038/s41590-021-01059-0 (2021).

Zhang, X. *et al.* *Nature immunology* **25**, 703–715; 10.1038/s41590-024-01782-4 (2024).

In summary, after thorough analysis of the Zhang and Triana datasets, we can confirm *ADGRG6* expression in the most immature cells in their subsets as well.

Letter Fig. 7. Gene expression of *ADGRG6* in most immature HSPC clusters from previously published studies. We analyzed *ADGRG6* expression from sequencing data of our study (Komic *et al.*), and of the studies by Zhang *et al.* (A) and Triana *et al.* (B)

Data and Code Availability: While I did see the data generated in a zenodo repository, it is crucial for the CITE-seq data to be published in an international repository prior to the acceptance of the manuscript. Similarly, could the authors clarify whether they have decided to release the analysis code openly, or will it remain restricted? I do see a github repo, yet the paper suggests code may not be made freely available? Restricting code would require further justification, as it significantly impacts the reproducibility and transparency of the study.

#Reply 5: At the time of primary submission to Nature Communications, we provided the single cell seq data via Zenodo using a restricted access for the editors and reviewers. Furthermore, we uploaded all codes to github repository (free access) on the 11th of May. However, the information about the access to Zenodo and the publication on github was accidentally not sent to the reviewers. We are sorry about the confusion and the inconvenience for the reviewers.

Because of the comment by the Reviewer, we decided to provide the primary sequencing data on the international repository ArrayExpress. Raw count matrices, sample tag calls, rds files and detailed sample information files can be accessed using the following link:

<https://www.ebi.ac.uk/biostudies/arrayexpress/studies/E-MTAB-14596?key=6eb62c54-2e2a-466e-bc78-a2c690645830>

Our R codes are openly available under:

<https://github.com/TessaSchm/Early-HSC-differentiation/tree/main>

Rebuttal letter with point-by-point replies to the reviewers' questions

First, we would like to thank the reviewers and editors for carefully reading and evaluating our revised manuscript, and for their most helpful comments. We are pleased to learn that our revised version has significantly improved, and that the reviewers acknowledged the value of our study for the scientific community. We thank Reviewer 2 for the acknowledgement of our rebuttal, and Reviewer 1 for pointing out important remaining questions which we all addressed in our newly revised version 2.

Reviewer #1 (Remarks to the Author):

The reviewer has some questions (see attached figure) about the now provided gating illustration of sorted CD273+ cells in Figure S9. There is a consensus on the immunophenotype of human HSCs: Lin-CD34+CD38-CD45RA-CD90+ and MPPs: Lin-CD34+CD38-CD45RA-CD90-. From the new S9a (now there are more cells), it is clear that the majority of CD273+ cells in BM or PB are CD90-, which falls more into the MPP immunophenotype.

The authors did not answer how CD273-high and low cells were gated for sorting (Q1), so I have to just assume it's based on the line I draw.

Reply: We apologize for not having shown the sorting gate of CD273-negative HSPCs in our old Suppl. Figure 9a. We thank the reviewer for investing the time to indicate missing information in the attached figure document. Since we decided - based on your recommendation - to present the FACS cell isolation data as **new main Figure 5**, we also included the FACS gates for sorting CD273^{high} and CD273^{low} HSPCs, according to the full-minus-one control. As you can see from this strategy, we avoided excluding many cells from functional analyses.

a

b

c

Figure 5. FACS sorting of CD273^{high} and CD273^{low} HSPCs

a, b) Dot-plots illustrating the hierarchical gating strategy used for isolation of CD34⁺CD38⁻CD45RA^{lin}⁻ HSPCs according to their CD273 expression from adult bone marrow (BM, a) and mobilized peripheral blood (mPB, b). The first plot shows pre-gated viable, singlet CD34⁺ lin⁻ cells. c) CD273 expression of CD34⁺ MACS-enriched mPB. d) Percentage of CD273^{high} cells within the indicated populations. *, p<0.05; **, p<0.01; n.s., not significant.

I'm also confused how CD34⁺CD38⁻CD45RA⁻CD90⁺ HSCs could have a higher MFI than CD34⁺CD38⁻CD45RA⁻ HSC&MPP because there are only 1–2 CD273-high HSCs (arrow pointed) in the flow (Q2).

Reply: Thanks for raising this critical point. We went back to our FACS data shown in **Figure 6a** and confirmed the findings in an extended data set, now analyzing n=11 samples, that the MFI of CD273 increases with immaturity of the HSPCs (from CD34⁺CD38⁻CD45RA⁻ to CD90⁺ CD34⁺CD38⁻CD45RA⁻ to CD49f⁺ CD90⁺ CD34⁺CD38⁻

CD45RA-). We agree with the reviewer that the example shown in **new Figure 5b** suggests a higher percentage of CD90- MPPs express CD273, however, the quantification of the expression level shows a significant increase of CD273 MFI in CD49f+CD90+ HSPCs (sorting scheme **new Supplementary Figure S11a**).

Extended dataset

Data shown in Figure 6a:

a

CD49f+, CD133+, EPCR+ gating (Fig. 6B) are missing, so make sure to provide representative flow plots whenever a gate is used.

Reply: Thank you very much for your suggestion, we included the gating of *in vitro*-cultivated HSPCs (Fig. 6b) in **new Suppl. Figure S11b**. We used several markers to determine the stage of differentiation in *in-vitro* culture.

Supplementary Figure S11. FACS gating of HSPC populations in functional analyses.
 a) Representative flow cytometry plots for analyzing CD273 expression levels on defined HSPC subpopulations with their percentages. b) Representative plots of *in vitro*-cultured CD273^{low} and CD273^{high} sorted HSPCs. Gates show CD34+ cells with expression of CD201(EPCR), CD90 and CD133 and their percentages.

The cell frequency/percentage of each gate should also be clearly labeled on the plots.

Reply: Thank you very much for this comment. We included the percentage of cells in each gate shown in the FACS plot representations of **new Figures 5** and **new Suppl. Figure S11**. → see figures above

Make sure to present flow data in a clear and transparent way because it is crucial to the major innovation of this paper.

Reply: Thank you for raising this important point. Therefore, we included flow cytometry plots and gating/sorting strategies for all data figures: Sorting for CD273^{high} and CD273^{low} HSPCs (**New Figure 5** referring to all functional analyses shown in Figures 6 and 7), gating for HSPC subpopulations (**Supplementary Figure S11a** referring to Figure 6a), and gating for *in-vitro* cultured HSPC differentiation stages (**Supplementary Figure S11b** referring to Figure 6b). → see figures above

The authors used the two terms HSCs (by their definition, CD34+CD38-CD45RA-Lin-) and HSPCs inconsistently in this paragraph, which is very confusing and should all be corrected to HSPCs.

Reply: We changed the term HSC to HSPC throughout the manuscript to elaborate on the remaining heterogeneity.

The flow illustration (with proper labeling) should be moved to the main figure because it is important to know whether this proposed marker can be used to sort enough cells (if readers can use it for their experiments).

Reply: Thank you for this suggestion. We show the flow cytometry data of CD273 expression in HSPCs now in a **new main Figure 5**. → see figures above

The subtitle of that paragraph might need to change because there are very few CD273+ cells in the most immature HSCs (CD90+), and validation was done on early HSPCs (mostly CD45RA- MPPs).

Reply: We changed the subtitles of the paragraphs showing the functional characteristics of CD273+ HSPCs accordingly. The new subtitles are “AbSeq reveals high expression of CD273 in immature HSPCs” (Line 290) and “CD273 functions as an immune-modulatory receptor in HSPCs” (Line 410)

These sorted CD273+ cells are a mixture of MPPs (90%) and HSCs (10%), are quiescent, and have the same lineage output. I think it is compelling that CD273 is a marker of quiescence among these early HSPCs; however, “most immature” HSPCs are stem cells, and there is no evidence of what impact CD273 has on CD90+ HSCs. Please avoid

making claims regarding “most immature” or “stem cell-like” in lines 52–53 to avoid confusion.

Reply: Thank you for this important point, we changed the wording in lines 52-53 to: “We identified CD273/PD-L2 to be highly expressed in a subfraction of immature multipotent HSPCs with enhanced quiescence.”

Regarding the new data in Figure 7, I’m not sure if it’s appropriate to use CD34+ HSPCs to prove CD273 protects immature HSPCs. CD34+CD38- is about 10–15% of CD34+ cells, CD34+CD38-CD45RA- is about 60% of CD34+CD38-, and CD34+CD38-CD45RA-CD273+ cells are 5% of CD34+CD38-CD45RA-. Therefore, out of the 10,000 CD34+ in co-cultures, there are probably just ~30 CD34+CD38-CD45RA-CD273+ cells. The data showed CD273 antibody blockade restored T cell activation in co-culture, and I found it hard to believe just 30 CD34+CD38-CD45RA-CD273+ cells could inhibit T cell proliferation.

Reply: We fully agree with the reviewer. Therefore, we avoided any claims in the results or discussion sections that these profound effects are specific to the most immature CD34+CD38-CD45RA-CD273+ HSPCs, but that CD273 blockade on CD34+ HSPCs show these effects.

In other words, are there other CD273+ cells within CD34+ compartments? This could easily be examined by comparing the number of CD34+CD273+ to CD34+CD38-CD45RA-CD273+ cells in the flow.

Reply: Thank you for this important question. We analyzed the percentage of CD273+ CD34+ HSPCs in mobilized peripheral blood samples, which we used for our mixed lymphocyte reaction, and added the data to **New Figure 5c and d**. In average, 1.3% of CD34+ HSPCs express CD273, which is a smaller fraction than in more immature HSPCs (about 5%). Based on these FACS results, about 100 CD273+ HSPCs were present in each coculture with T cells.

We added following sentence to the results (line 415): “In average, about 1% of HSPCs expressed CD273 on their surface (Figure 5c and d).”

New Figure 5. ... c) CD273 expression of CD34⁺ MACS-enriched mPB. d) Percentage of CD273^{high} cells within the indicated populations. *, p<0.05; **, p<0.01; n.s., not significant.

Or the authors could sort CD34+CD38-CD45RA-CD273+/high cells vs. CD34+CD38-CD45RA-CD273-/low cells and test their immunomodulatory effect on immune cells in co-culture.

Reply: As originally suggested by the reviewer, blocking experiments using a validated anti-PD-L2 antibody, has several advantages in functional analyses: first, CD34+CD38-CD45RA-CD273-/low cells upregulate CD273 expression upon *in vitro* culture within a few days. Second, FACS sorting can be avoided to prevent cell stress and activation. Third, technical sorting impurity does not influence the results. Fourth, antibody blockade is long-lasting and highly selective. This was an excellent recommendation by the Reviewer, therefore, we showed the effects of T cell proliferation and activation upon blocking CD273.

The authors have shown advantages of their approach and dataset over the two existing atlases respectively: 1) More early CD34+ HSPCs than Triana/Haas'; 2) Targeted Ab-seq provides a better profiling of low-expression genes such as MPL compared to Zhang/Grimes'. It would be useful to provide a list of these "missed" genes, including the ones provided in the rebuttal letter, in the supplementary table and mention them in the discussion.

Reply: We included a complete list of genes identified as markers for our HSC-1 cluster and compared their presence in the data sets of Triana *et al.* and Zhang *et al.*, shown as **Supplementary table S8**. We further graphically represented the expression data of these genes in the most immature HSPC clusters of each dataset (examples) in the **new Supplementary Figure S7**.

We added following sentence to the results (lines 182ff): "We confirmed increased expression of *DLK1* and *ADGRG6* in the most immature HSPC clusters of previously published single cell expression datasets, whereas some HSC-1-specific genes where

not differentially detected in these datasets (Supplementary Figure S7, Supplementary Table 8).”

Supplementary Figure S7. Comparison of HSC-1 upregulated genes.

Gene expression of HSC-1 up-regulated genes identified in our study (Komic *et al.*) and compared to the expression in the most immature clusters of the previous studies of Zhang *et al.*, 2024 and Triana *et al.*, 2021. *, $p < 0.01$; **, $p < 0.001$; ***, $p < 0.0001$; ns, not significant.

Use Abseq, not CITE-seq. BD already trademarked Abseq to make it their own thing to compete against CITE-seq and Biogend Totalseq. BD would much prefer you use Abseq instead of CITE-seq. As for “searchable,” Triana/Haas used Abseq consistently (never mentioned CITE-seq) in their 2016 Nature Immunology paper, and it has 54k accesses and an altmetric score of 245.

Reply: We changed the term CITE-Seq to “Transcriptomic/AbSeq” throughout the manuscript.

The added CD273 function data/research direction has now better strengthened the biological findings, making it the highlight of this study. CD273 is a new immunomodulatory marker that helps early HSPCs. Since there is no major innovation in analyzing the data, the reviewer suggests limiting the paragraph and figures describing markers and unvalidated hypotheses (transcriptome clusters/trajectories without functional isolation and validation) that are just consistent with the literature. Some suggestions as below.

Reply: We followed your recommendations and have significantly shortened these sections in the results and discussion. We further moved **Figure 4** to the supplementary figures.

We shortened the Results section to 8 pages, and the Discussion to 2 pages.

The revised abstract, line 49: change “demonstrate” to “suggest.”

Reply: We changed the word “demonstrate” to “suggest”.

Trajectory and pseudotime analysis without functional validation (e.g., cell tag tracing or captures at different time points or index sort) do not have the ability to demonstrate or prove anything. Line 55: remove “comprehensive” per discussion before.

Reply: We removed “comprehensive”.

Notta/John Dick already have that concept of Mk/E branching from early HSC/MPP in their 2016 Science paper. The “CITE-seq of CD34+ HSPCs reveals early Mk/E lineage branching” paragraph (as the first and main claim) provides nothing conceptually new but more of a technical description of the dataset, and most of it can be moved to the method.

Reply: We changed the subtitle of the first paragraph of the results to: “Reconstructing a hierarchical organization of CD34⁺ HSPCs by targeted Transcriptomic/AbSeq”. We further added the citation of Notta *et al.* Science 2026 to the statement (Line 139).

This whole paragraph simply shows manually annotated HSC/MPP are transcriptomic stem and progenitors.

Reply: We feel that it is of importance to demonstrate the accuracy and applicability of our workflow to the readership, therefore the first paragraph of the manuscript contains primarily many important control experiments to show the validity of our analyses and annotations.

Myeloid bias and increased HSPCs in old people are well-known.

Reply: We shortened the statements to a minimum, and do not claim any original findings. However, we also feel that it remains important to demonstrate that our data follows the reported age-associated changes.

Figure 4 provides little readable information and can be moved to the supplemental.

Reply: We moved **Figure 4** to the supplements and created a **new Supplementary Figure S9**. Furthermore, we extracted the tradeSeq data of four HSC-1 specific genes (from **old Figure 4**), and included these plots into **New Figure 3f**.

The discussion is too long; just focus on biological findings and avoid too much speculation. Lines 385–396: Although I agree and believe hematopoiesis is a stepwise commitment rather than a “cloud,” it takes a lot more to claim these discrete cell states, such as by isolating transcriptomic cell populations and functionally validating them.

Reply: Thank you for this valuable suggestion. We restructured the discussion and drastically shortened it to 2 pages. We put most focus on the findings of CD273 expression on HSPCs and its functional role to modulate T cell activation. We deleted or shortened the discussion of other findings of our study to a minimum, as suggested by the Reviewer.

The new discussion reads as follows:

Discussion

In this study, we investigated molecular cues associated with human HSPC identity, early differentiation induction and lineage choice using single cell proteo-genomic sequencing. The targeted sequencing approach allowed accurate detection of weakly expressed genes at single cell resolution of CD34⁺ BM cells across different age groups for continuous gene expression quantification. The combination of surface marker detection with the single cell transcriptome enabled direct detection of new surface markers with differential expression in the most immature HSPC fraction.

Thereby, we identified CD273/PD-L2 as a differentially expressed surface marker in human HSPCs. CD273 is a ligand of PD-1 and plays an important role as an immunomodulatory molecule. The inhibition of the PD-1/PL-L1/L2 axis is highly relevant in cancer treatment, known as immune checkpoint blockade⁴⁶. Activated PD-1-expressing T-cells are suppressed by the interaction with PD-L1 or L2, which are commonly overexpressed on cancer cells or myeloid cells in many solid malignancies⁴⁷. In murine hematopoiesis, the expression of CD274 (PD-L1) was reported to allow murine HSPCs to escape immune reactions mediated by HSPC-reactive T-cells^{48,49}. HSPCs have been proposed to have immune modulatory functions due to expression of MHC class II molecules that may allow HSPCs to present antigens influencing T-cell responses⁵⁰. Lately, Beta-2-microglobulin expression on HSPCs has been proposed as a crucial "don't eat-me" signal in zebrafish, regulating the HSPC-macrophage interaction and HSPC clonality.⁵¹ However, the expression of PD-1 ligands may serve other purposes than protecting HSPCs against auto-reactive T-cells. In our study, we could confirm a higher surface expression of CD273/PD-L2 on a subset of human immature HSPCs by flow cytometry. Prospectively isolated HSPCs with high CD273 expression were found to display delayed differentiation, multi-lineage potential and enhanced quiescence, all attributes important for HSPC physiology. Increased expression of stemness genes suggests that potent human HSPCs may reside in the CD273^{high} HSPC fraction, which needs adequate functional *in vivo* assessment. Here we demonstrated an immune-modulatory function of CD273/PD-L2 expressed by HSPCs. Allogeneic T-cell proliferation and activation were elevated upon CD273/PD-L2 blockade. Furthermore, the release of pro-inflammatory cytokines increased. Lastly, T-cell differentiation into regulatory T-cells (T_{reg}) was reduced after CD273/PD-L2 inhibition on HSPCs. These results suggest that CD273/PD-L2 plays an immunosuppressive role on HSPCs, preventing excessive T-cell activation and potential auto-reactivity. The data highlight CD273/PD-L2 as a safeguard for immature HSPC populations, acting as both a direct inhibitor of T-cell responses and as a facilitator of T_{reg}-mediated immune regulation. The presence of regulatory T-cells within the HSPC BM niche may provide a privileged environment for HSPC physiology.^{52,53} These findings provide valuable insights into the complex immune interactions between HSPCs and T-cells, and underscore the importance of CD273/PD-L2 in maintaining immune homeostasis.

In agreement with recent reports of single cell sequencing studies using human BM cells and cord blood cells, we here present a structure of a hierarchical organization of human hematopoiesis^{11,32,54}. Initial single cell sequencing data of human BM suggested that there is a "cloud" of HSPCs, which very rapidly become committed to a distinct lineage.⁹ Our data provide evidence for a model of stepwise commitment and restriction to distinct lineages, which would be compatible with decades-long findings of the murine hematopoietic hierarchy. There was a considerable delay in pseudotime, before the most immature HSPCs at the anchor point of the trajectories diverged into the first lineage-committed progenitors (Mek/E lineage) suggesting that multipotency is maintained after the first steps of early differentiation. While we have molecular evidence that most immature HSPCs are multipotent, rather than a heterogeneous mixture of lineage-restricted HSPCs, we did not perform single cell transplantations to consolidate their multipotency. Pseudotime and tradeSeq analyses suggested a balanced and productive differentiation of HSPCs from young donors, and a delay in differentiation of all lineages in mid-age and old donors by an increased expression of stemness genes. This

diminished differentiation was most pronounced in the lymphoid commitment. These findings suggest that there is an age-dependent molecular regulation of progenitor production by early events in HSPC differentiation, contributing to the increased frequency of HSPCs as well as the reduced lymphopoiesis and reduced normal range of hemoglobin seen in old people. Whether these changes were caused intrinsically or orchestrated by the aged environment requires further investigation.

A major strength of our study is the accurate quantification of even lowly expressed genes at single cell resolution by targeted sequencing with a low individual dropout rate, which distinguishes our study from previous attempts. However, this strength naturally comes with the tradeoff that our analyses were restricted to a set of preselected protein-coding genes. Nevertheless, our data demonstrates that the selection of these genes was sufficient to clearly resolve human HSPCs populations with similar efficiency as compared to whole transcriptome analysis.

In conclusion, here we present a rich data set of single cell gene expression changes influencing the earliest steps of human HSPC differentiation. Using our approach, we discovered that the immune-modulatory molecule CD273/PD-L2 is highly expressed on the most immature HSPCs and its functional capacity in suppressing allogeneic T-cell proliferation, activation and pro-inflammatory cytokine release. These findings have implications on our understanding of human HSPC physiology and differentiation control, important features for therapeutic manipulation and expansion of human HSPCs.

Reviewer #1 (Remarks on code availability):

Data and code transparency is greatly improved.

Reply: Thank you very much.

Reviewer #2 (Remarks to the Author):

The authors have carefully responded to the reviewer questions. I am happy with the revised version.

We would like to thank the reviewer for the critical evaluation of our study, and the positive and helpful comments to improve the manuscript.

Figure S9a

a

Q1: how did you define CD273 low and high?

CD273low? CD273hi?

Figure 6a

Q2: CD34+CD38-CD45RA-CD90+ cells have higher MFI than CD34+CD38-CD45RA-?